# Comparative analysis of the unbinding pathways of antiviral drug Indinavir from HIV and HTLV1 proteases by supervised molecular dynamics simulation

**Farzin Sohraby** [ID] [c], **Hassan Aryapour** [ID] *[c]

Faculty of Science, Department of Biology, Golestan University, Gorgan, Iran

[c] These authors contributed equally to this work.
* h.aryapour@gu.ac.ir

## Abstract

Determining the unbinding pathways of potential small molecule compounds from their target proteins is of great significance for designing efficacious treatment solutions. One of these potential compounds is the approved HIV-1 protease inhibitor, Indinavir, which has a weak effect on the HTLV-1 protease. In this work, by employing the SuMD method, we reconstructed the unbinding pathways of Indinavir from HIV and HTLV-1 proteases to compare and understand the mechanism of the unbinding and to discover the reasons for the lack of inhibitory activity of Indinavir against the HTLV-1 protease. We achieved multiple unbinding events from both HIV and HTLV-1 proteases in which the RMSD values of Indinavir reached over 40 Å. Also, we found that the mobility and fluctuations of the flap region are higher in the HTLV-1 protease, making the drug less stable. We realized that critically positioned aromatic residues such as Trp98/Trp98′ and Phe67/Phe67′ in the HTLV-1 protease could make strong π-Stacking interactions with Indinavir in the unbinding pathway, which are unfavorable for the stability of Indinavir in the active site. The details found in this study can make a reasonable explanation for the lack of inhibitory activity of this drug against HTLV-1 protease. We believe the details discovered in this work can help design more effective and selective inhibitors for the HTLV-1 protease.

## Introduction

In the last decades, retroviruses have always been a significant threat to human beings′ lives, and since then, countless people have lost their lives as a result. Retroviruses are the oldest family of viruses on planet earth. Recent bioinformatics advances have estimated that retroviruses have originated about 500 million years ago [1] and were inserted into the human genome about a million years ago [2]. Over these years, the genome of retroviruses, like most families of viruses, has been added to our genomes, and as a result, about 8 to 10% of the human genome has a viral origin which is the sign of ancient germ line infections. They are known as

**Data Availability Statement:** The data underlying this study are available on Zenodo (https://zenodo.org/record/5121095#.YPpq470zblU).

**Funding:** This study was supported by Golestan University, Gorgan, Iran.

**Competing interests:** The authors have declared that no competing interests exist.

Endogenous Retroviruses (ERVs) [1,3], which may have probably evolved from transposable elements. However, RNA-enveloped Viruses such as Human Immunodeficiency Virus (HIV) or Human T-cell Leukemia Virus 1 (HTLV-1) have to be transmitted horizontally among hosts and belong to the subfamily of Exogenous Retroviruses (XRVs).

HIV has been one of the dangerous viruses on planet earth. Since its occurrence in the 1980s, this lethal virus has taken millions of lives. Since the start of the epidemic, 32 million people have died from AIDS-related illnesses [4]. This pandemic virus forced health organizations to speed up the research to find treatment options. Fortunately, the extensive research and development to eradicate HIV led to treatment solutions with good efficacy. In 1987, the FDA approved the first drug to treat HIV, called Zidovudine (AZT). AZT belongs to a class of drugs known as Nucleoside Reverse Transcriptase Inhibitors (NRTIs), which cause premature determination of the proviral DNA. Many more drugs and solutions were achieved afterward, and now HIV-infected individuals are almost able to live everyday life with a usual life expectancy.

The viral proteases are one of the critical targets for the treatment of virus infection. Over the last 20 years, HIV protease inhibitors have shown that inhibiting viral protease can be an excellent strategy to fight the virus. Nowadays, protease inhibitors are one of the main parts of the combination therapies of HIV treatments. Approved drugs such as Indinavir, Saquinavir, etc., have good efficacy for inhibiting the HIV protease and have been proven to have a significant impact on the condition of the patients [5,6]. The first Anti-Retroviral Therapy (ART) was based on NRTIs, which were not very effective and caused many complications, but the development and the emergence of protease inhibitors introduced a new ART milestone. This milestone gave hope to both the developers and also the patients that this infection may be conquerable. Combination therapies provide excellent efficacy and show that protease inhibitors are the main parts of successful treatments [7–12].

The HTLV-1 virus is associated with adult T-cell leukemia (ATL) and an inflammatory disease syndrome called HTLV-1-associated myelopathy/tropical spastic paraparesis (HAM/TSP) [13–15]. About 90 percent of infected individuals remain asymptomatic, but due to the virus′s permanent entanglement with the immune system, the infected person will suffer from immune system deficiencies throughout his/her life, leading to other problematic diseases [16–18]. HTLV-1 has infected about 20 million people worldwide [15], and treatment solutions are urgently needed. Anti-ATL vaccines have been developed recently and are in clinical trials, but they are designed to fight the cancerous cells, not the virus itself [19]. In contrast to HIV, which has first-line therapies and treatments are available for it, HTLV-1 does not have any treatments to this date. A solid strategy to fight this virus is the direct inhibition of its essential work machines, such as the protease enzyme, the same as the effective HIV treatments [20]. As mentioned before, protease inhibitors have proved their efficacy, and in the case of HTLV-1, protease inhibitors may be a potential breakthrough in HTLV-1 treatments [21].

In the last few years, all of the inhibitors of the HIV protease have been tested against the HTLV-1 protease. However, none of the inhibitors exhibited sufficient inhibition. Although the two enzymes are very similar in structure, the anti-HIV drug with an inhibitory constant (Ki) of 540 pM [22], Indinavir, could only achieve an inhibitory constant of 3.5 μM against the HTLV-1 protease [23,24]. The only consolation is that the cause of this can only be the difference in their structural behavior and the difference in the sequences of these two enzymes. The residues of the active site and the flap regions can especially dictate this behavior [25–28].

Computational methods such as Molecular Dynamics (MD) simulation enables researchers to thoroughly study the structural details of complex mechanisms of biomacromolecules [29]. Studying the unbinding pathway of small molecule compounds in complex with their target

proteins is a great feature of the MD simulation method that has been made possible in the last decade. Many research groups developed great approaches over these years that empower us to understand the fine details of these mechanisms [30–44]. Unraveling the drug′s unbinding pathway from its target protein is an excellent task that can be very useful for understanding the mechanism of the unbinding process, as well as the essential residues involved [45]. The details achieved can be used to make more effective and more selective drugs. One of the appropriate approaches for this task is the Supervised MD (SuMD) simulations [41]. In this approach, the simulation of the protein-ligand complex is performed in replicas with fixed duration times, and at the end of each replica, a specific parameter such as the distance of the ligand from the binding site is checked, and the frame with the highest distance is extracted and extended as the next replica. This cycle will be continued until the ligand is entirely unbound from the target protein. This atomistic approach is entirely unbiased, and there are no artificial attractive or repulsive forces involved. Also, similar out-of-equilibrium approaches such as weighted ensemble milestoning (WEM) and forward flux sampling (FFS) have been developed so far that have been used for ligand unbinding processes [46,47]. Herein, by utilizing the SuMD approach under equilibration conditions, we tried to achieve the anti-HIV drug′s unbinding pathway, Indinavir, from both the HIV protease and the HTLV-1 protease as a comparison test to understand the details and the behavior of these critical protein targets. This strategy can be effective in finding the reasons for the lack of efficacy of the HIV protease inhibitors against the HTLV-1 protease.

## Methods

The X-ray crystallography structures of the HTLV-1 (PDB ID: 3WSJ) [24] protease and the HIV protease (PDB ID: 1K6C) [48] in complex with Indinavir were obtained from the Protein Data Bank [49]. The structures were prepared by UCSF Chimera software [50]. All of the unnecessary molecules, such as water molecules, were deleted from the structures, and the protein-ligand complexes′ structures were ready for the next step, the MD simulations. There are two catalytic Asp residues in both proteases′ active sites, and one of them must be protonated during the simulation. The extra hydrogen atom was added to the Asp residue of chain A in both cases. All of the MD simulations were done by GROMACS 2018 package [51] and OPL-S-AA force field [52]. The 3D structures of the co-crystallized Indinavir were parameterized using ACEPYPE [53] with the default setting for assigning the partial charges and atom types. For the construction of the simulation systems, first, the related protein-ligand complex was placed in the center of a triclinic box with a distance of 1 nm from all edges and then solvated with the TIP3P water model [54]. About 9400 and 7300 water molecules were added to the HTLV-1 and HIV protease systems, respectively. Then, sodium and chloride ions were added to produce a neutral physiological salt concentration of 150 mM. Each system was energy minimized, using the steepest descent algorithm, until the Fmax was smaller than 10 kJ.mol$^{-1}$.nm$^{-1}$. All of the covalent bonds were constrained using the Linear Constraint Solver (LINCS) algorithm [55] to maintain constant bond lengths. The long-range electrostatic interactions were treated using the Particle Mesh Ewald (PME) method [56], and the cut-off radii for Coulomb and Van der Waals short-range interactions were set to 0.9 nm for the interaction of the protein-ligand complex. 100 ps of NVT and 300 of NPT were performed using the modified Berendsen (V-rescale) thermostat [57] and Parrinello–Rahman barostat [58], respectively, for the equilibrations and to keep the system in stable environmental conditions (310 K, 1 Bar) during the production runs. Finally, SuMD simulations [59] were carried out with a time step of 2 fs. The periodic boundary condition (PBC) was set at XYZ coordinates to ensure that the atoms had stayed inside the simulation box during the equilibration and production runs. The

subsequent analyses were then performed using GROMACS utilities, VMD [60] and USCF Chimera, and also the plots were created using Daniel′s XL Toolbox (v 7.3.2) add-in [61]. The free energy landscapes were rendered using Matplotlib [62]. In addition, to estimate the interaction energies, we used the g_mmpbsa package [63]. The Free Energy Landscape (FEL) analysis was done by "gmx sham" module of GROMACS software.

The difference between the conventional MD (cMD) and the SuMD simulations is the fact that SuMD is an out-of-equilibrium simulation and, also in SuMD, the entire simulation is divided into a series of replicas, and a specific parameter is monitored throughout them as the guideline to choose the starting point of the next replica. The original idea was proposed by Giuseppe Deganutti et al. [41], which is an excellent methodology for achieving unbinding events. In the original method, "A series of short unbiased MD simulations are performed, and after each simulation, the distances (collected at regular time intervals) are fitted to a linear function. If the resulting slope is negative (showing progress toward the target), the next simulation step starts from the last set of coordinates and velocities produced; otherwise, the simulation is restarted by randomly assigning the atomic velocities", whereas, in our series of replicas, the procedure is much simpler. We considered the distance between the Center Of Mass (COM) of all of the atoms of the drug and the COM of the entire atoms of the Asp32 (chain A) and Asp32′ (chain B) in the HTLV-1 MD system and the Asp25 (chain A) and the Asp25′ (chain B) in the HIV proteases MD system as the guideline for selecting the best frame to be the starting point of the next replica. The distance between the COM of Indinavir and the mentioned residues in each system was monitored in the entire simulation. The duration time of all of the replicas in this study was set to 500 ps. At the end of every 500 ps simulation, the frame with the highest distance was selected as the next 500 ps simulation starting point. This procedure was done automatically by an external python script. We also performed 50 ps of NVT and NPT equilibration runs after the frame selection before the 500 ps production runs. This methodology addressed the out-of-equilibration problem of SuMD simulation.

## Results and discussion

In this study, by utilizing SuMD simulations, we tried to reveal the unbinding pathway of the anti-HIV drug, Indinavir, from the HIV and the HTLV-1 proteases to find the details of the unbinding mechanism and also to compare the results to understand the lack of inhibitory activity of this drug against the HTLV-1 protease.

The 3D structure of the HTLV-1 and the HIV proteases are very similar. When superimposed, the backbone RMSD value is as low as 0.95 Å (Fig 1A). Although the positions of the backbone atoms and the protein fold are roughly the same, the sequence identity falls to 28 percent [64]. The HTLV-1 protease has 125 residues, whereas the HIV protease has 99 residues. However, the conserved residues sequence in the active sites has an identity value of 45 percent [64]. The various regions of these two enzymes have high structural similarities since their function are the same. However, the details of the two enzymes' active sites differ in some points to recognize and accommodate their specific substrates. As a result, none of the HIV protease inhibitors, except Indinavir, can inhibit this enzyme with relatively low concentrations. Kuhnert et al. determined and compared the structures of HIV and HTLV-1 proteases in complex with Indinavir and characterized the role of residues inside the active sites, and discovered significant deviations for the interaction networks of each moiety (Fig 1B) with the important sections of the binding pocket [24].

We achieved three unbinding events from each of the complexes in three series of replicas. The replicas were continued until the distance between the two centroids (COMs) reached roughly over 4 nm, and the ligand was in the unbound conformation. In this work, the

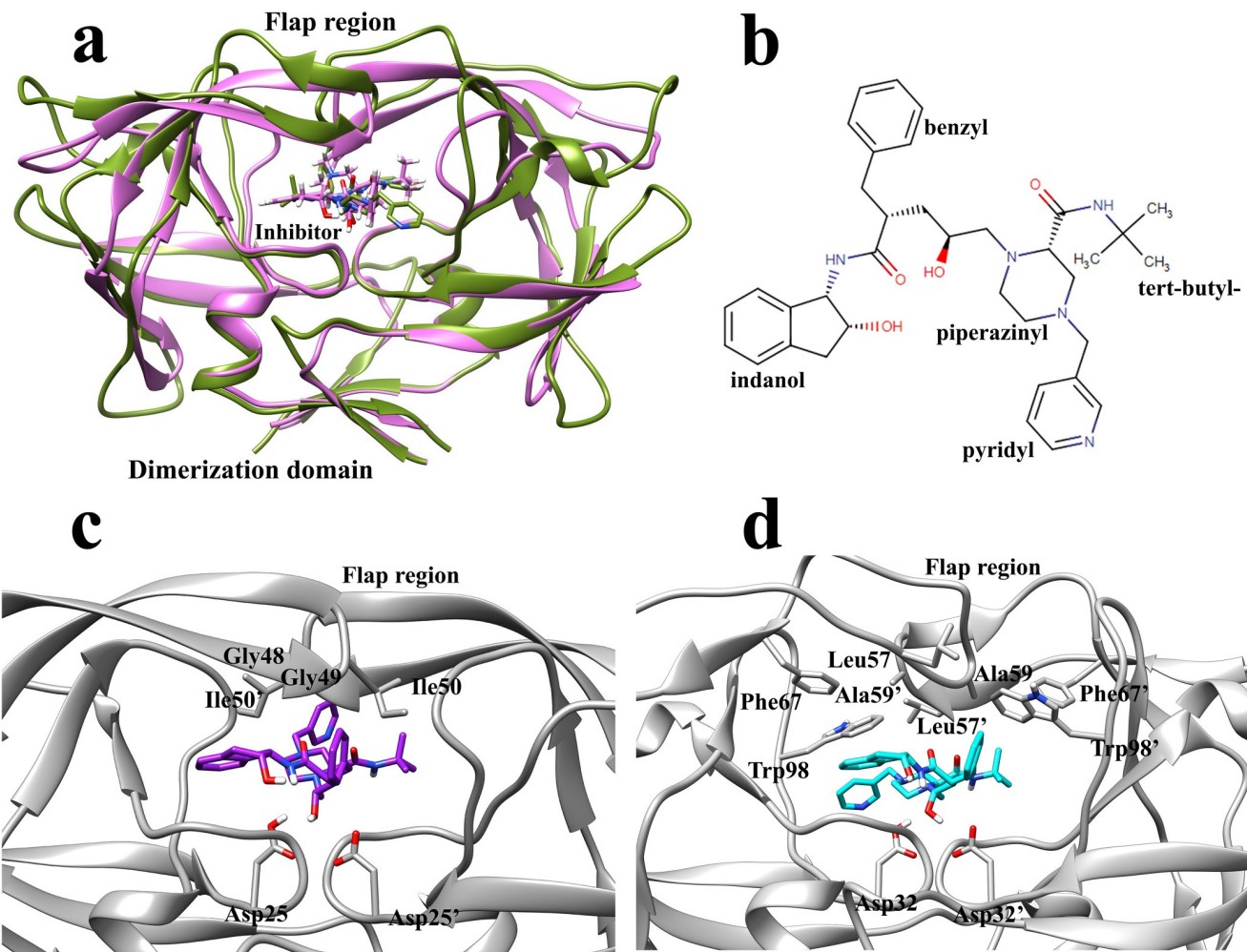

**Fig 1. Structure of HTLV-1 and HIV protease. a,** The superimposed structure of HTLV-1 (green) and HIV (pink) protease in complex with the inhibitor, Indinavir, is located in the two enzymes' active site. The backbone RMSD value was calculated by UCSF chimera software. **b,** The 2D structure of Indinavir and its corresponding moieties and substitutes. **c,** The position of Indinavir and the residues in the active site of HIV protease. **d,** The position of Indinavir and the residues in the active site of HTLV-1 protease.

unbinding events happened in 173, 183.5, and 338 ns in the HIV protease-Indinavir complex (Fig 2A and S4–S6 Files) and the HTLV-1 protease-Indinavir complex; the unbinding events happened in 243.5, 552.5, and 671 ns (Fig 2B and S1–S3 Files). Although Indinavir inhibits HIV protease much more selectively than the HTLV-1 protease, the overall time needed for the unbinding events of the HTLV-1 protease case was considerably more than that of the HIV protease case. However, many factors govern an inhibitor's activity, and only comparing the overall duration of the unbinding events in three series of replicas is not a correct way of comparing these two cases. This may be due to the inherent limitations of the SuMD method. Because the time window is set to 500 ps, therefore more conformational sampling by ligand is limited.

There are no biasing forces involved in the simulations, and they are entirely unbiased. The main difference between this method and conventional MD simulations is the automatic supervision at choosing the most appropriate frame in a replica for extending the simulation and also the fact that SuMD is an out-of-equilibrium simulation. As explained in the methods

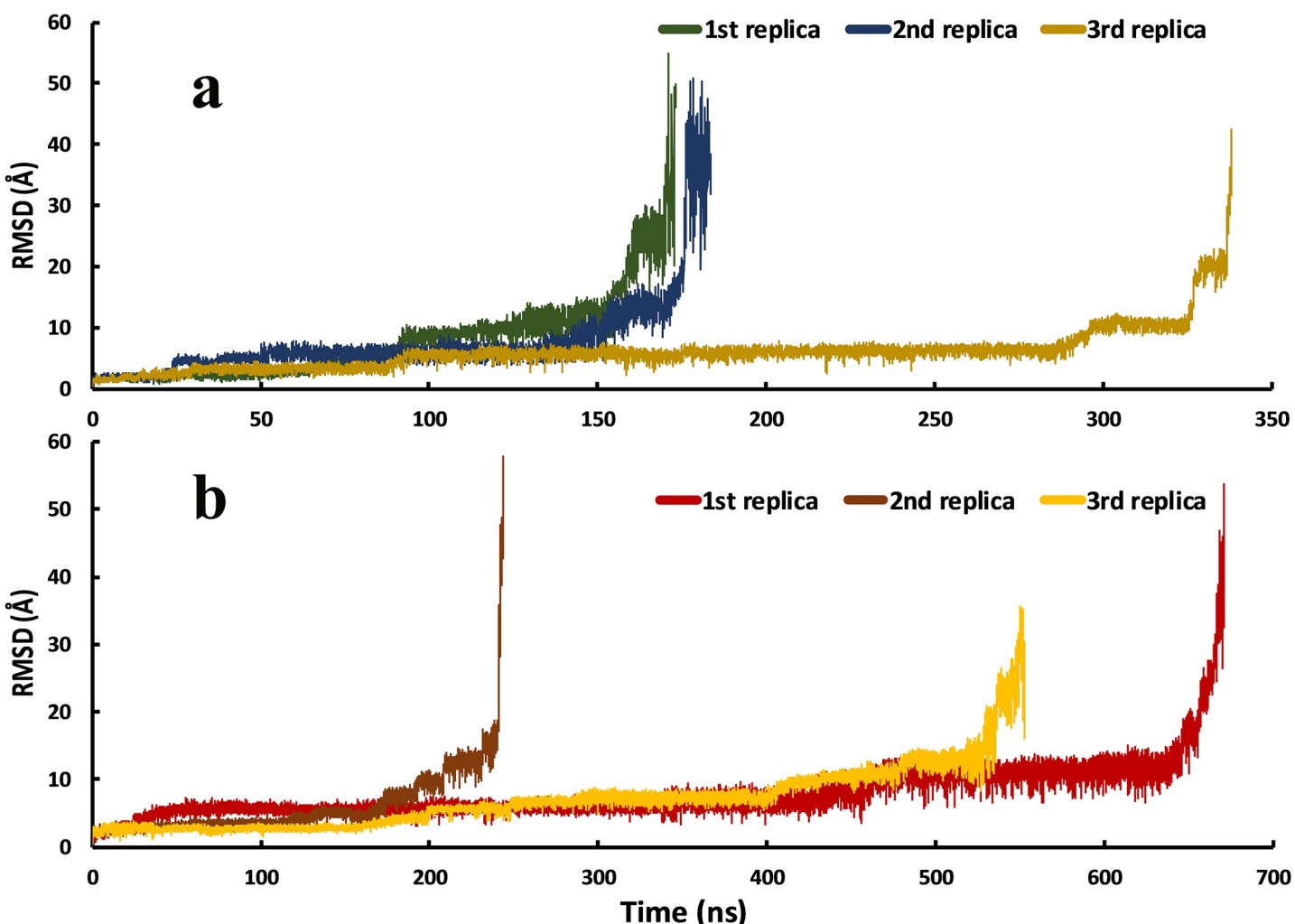

**Fig 2. The RMSD values of the Indinavir in the unbinding pathways in the three series of replicas for each protein-ligand complex. a,** Indinavir-HIV protease complex and **b,** Indinavir-HTLV-1 protease complex. Values were calculated using the crystallographic binding pose as the reference.

section, the most appropriate frame in a 500 ps simulation was the frame with the highest distance between the ligand and the catalytic Asp residues (S1 Fig). The SuMD approach is speedy and a remarkable tool for reconstructing the unbinding pathways of small molecule drugs and unraveling their details which are of great significance in drug discovery and design.

The unbinding pathway of Indinavir from the HTLV-1 protease and the HIV protease was very similar. In total, three stable states were observed during the unbinding pathways; (i) the Native state (N), (ii) the Intermediate states ($I_1$, $I_2$) and, (iii) the Solvated state (S) (Fig 3A–3F). The native state is where the ligand is in the crystallographic conformation. Indinavir can stay in this state for a long time in the unbinding pathway. In this state, the contact surfaces between the ligand and the protein are at their maximum, and the RMSD values are at their minimum. The next state in the unbinding pathway is the intermediate state $I_1$ and $I_2$, where the ligand has a tight interaction with the flap regions' residues. The two flaps in both proteases have high mobility, and their strong interaction with the ligand can make it get out of the crystallographic conformation in which it has fewer contact surfaces with the protein and higher RMSD values. This is the $I_1$ state. These intermediate states of Indinavir in the unbinding

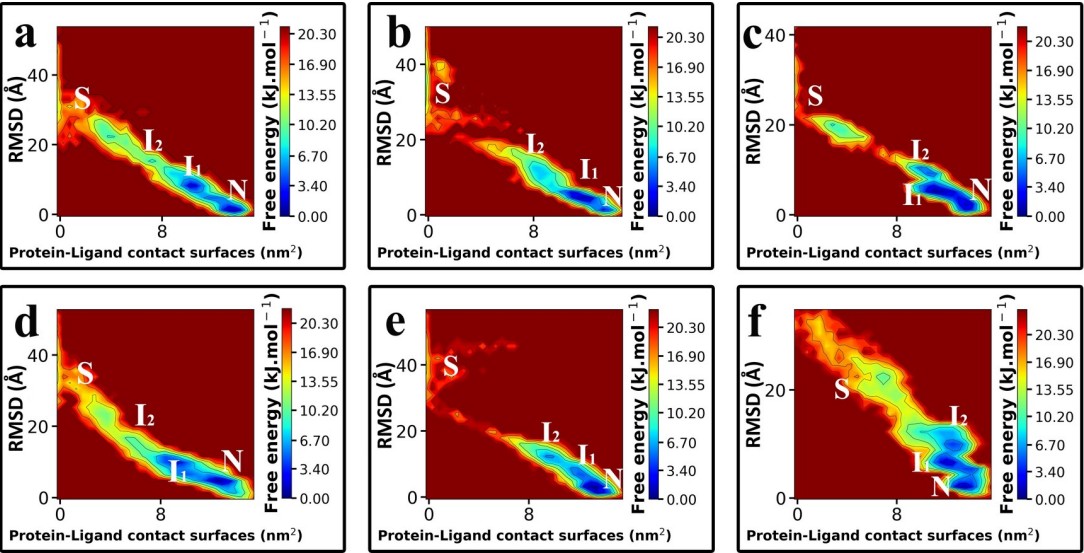

**Fig 3. The "out-of-equilibrium" Free Energy Landscapes (FEL) of the unbinding pathways of Indinavir. a, b, c**, from the HIV protease and, **d, e, f**, from the HTLV-1 protease. The stable states of Indinavir during the unbinding pathway are indicated by capital letters; Native (N), Intermediate ($I_1$, $I_2$), and Solvated (S).

pathway are stable, and the ligand can stay in them for long times. After the flap region was opened, the ligand can get attached to one of the flaps and get further from the active site, and this is the $I_2$ state.

The total interaction energies of Indinavir in the unbinding pathways of both proteases, calculated by the MM/PBSA approach (Fig 4A and 4B), showed roughly the same amounts. This means that although the residues of these two enzymes' active sites only have 45 percent identity, the interaction profiles are almost the same [23]. The values gradually increased and eventually reached zero when the ligand left the binding pocket and got solvated. Moreover, this analysis showed that the dominant interactions are the short-range VdW interactions between the aromatic and aliphatic residues and the ligand (S2 Fig). This finding indicates that the binding pockets of these two enzymes are somewhat hydrophobic. Except for the two Asp residues in the active site's deepest part, almost none of the residues can make direct hydrogen bonds with the ligand using their side chains. The hydrogen bonds are mainly formed with the residues' backbone by incorporating water molecules [24]. There is also a conserved water molecule in the Indinavir-HTLV-1 protease complex known as the "flap water," which mediates interactions between the ligand and Ala59 on the tip of the flaps (see Ref 24 for more information) [24].

As mentioned above, the interactions between Indinavir and the residues of both enzymes' binding pockets have almost the same amount and nature. However, each enzyme's active site residues' contribution to the total interaction energies differs in some points (Figs 5A, 5B and S3). In the Indinavir-HIV protease complex, Ile50, Gly49, Gly48 from the flap region, and Asp25 and Thr80 had the most contribution in the interaction energies (Fig 1C). In the Indinavir-HTLV-1 protease complex, Leu57, Gly58, and Ala59 from the flap region and the Trp98 and Asp32 in the active site had the most contribution in the interaction energies (Fig 1D). These emphasize the vital role of the residues of the flap region. It is also clear that in the HIV protease, almost all of the critical residues are aliphatic. However, in the HTLV-1, aromatic residues such as Trp98 and Phe67 conceal the role of aliphatic residues in the unbinding pathway. The existence of these aromatic residues in the active site of the HTLV-1 protease is one

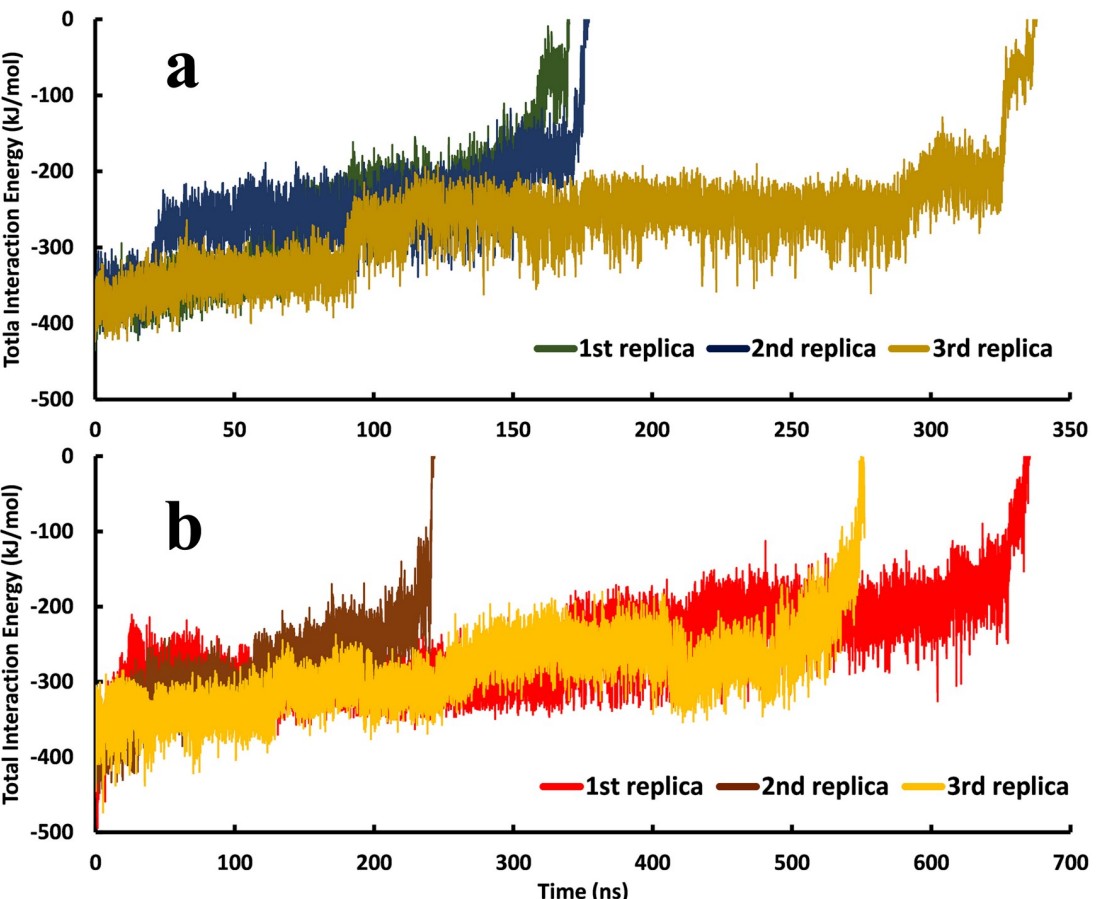

**Fig 4. The total interaction energies of Indinavir in the unbinding pathways in the three series of replicas for each protein-ligand complex. a,** Indinavir-HIV protease complex. **b,** Indinavir-HTLV-1 protease complex.

of the main reasons for the lack of proper inhibitory activity of Indinavir against this enzyme, which will be discussed later [24].

In this study, we also found that one of the most important structural differences between the two proteases is the level of mobility of the flap region. In both enzymes, this region has high mobility, but in the HTLV-1 protease, the flexibility and the fluctuations of flaps are slightly more than that of the HIV protease due to the nature of residues and the larger length of the flaps (Fig 6A and 6B) [65–68]. The higher mobility of this region is unfavorable for Indinavir to stay stable in the active site. In the presence of a ligand inside of aspartic proteases, the fluctuation of the flap region is significantly decreased, which makes the ligand very stable in the active site [69,70]. However, in the Indinavir-HTLV-1 protease complex, the higher mobility and fluctuation of this region might play a big part in the lack of sufficient inhibitory activity of Indinavir. The handedness feature of the flaps was also visible in the RMSF values in which one of the flaps has more fluctuation than the other [67,71].

Furthermore, in almost all three replicas of the Indinavir-HTLV-1 protease complex, the RMSD values of the flap region are noticeably higher than that of the HIV protease (Fig 6C and 6D). Ala59 in the HTLV-1 protease and Ile50 in HIV protease is present on the tip of the flap region. The Ile50 on the tip of the flap region in HIV protease may affect keeping the flaps in the closed state due to its strong interaction with Indinavir (Figs 6A and 7A). On the other hand, in HTLV-1 protease, Ala59 is present on the tip of the flaps, making weaker interactions

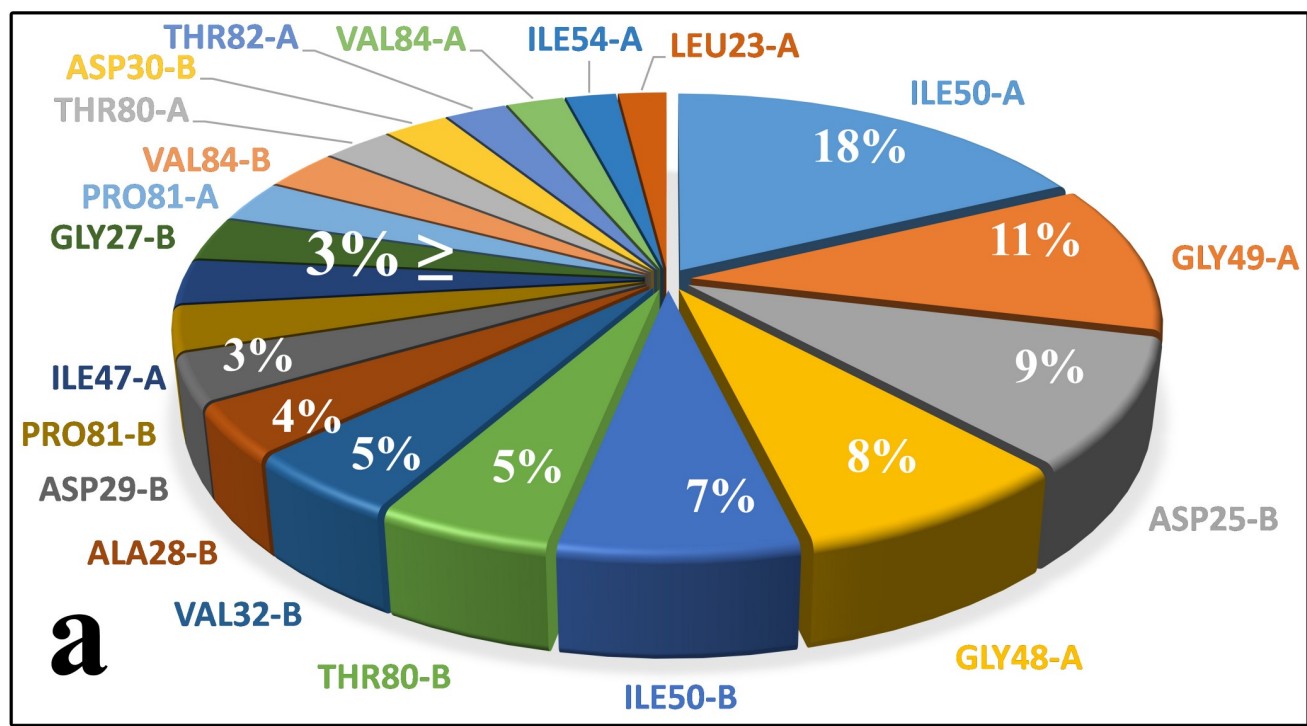

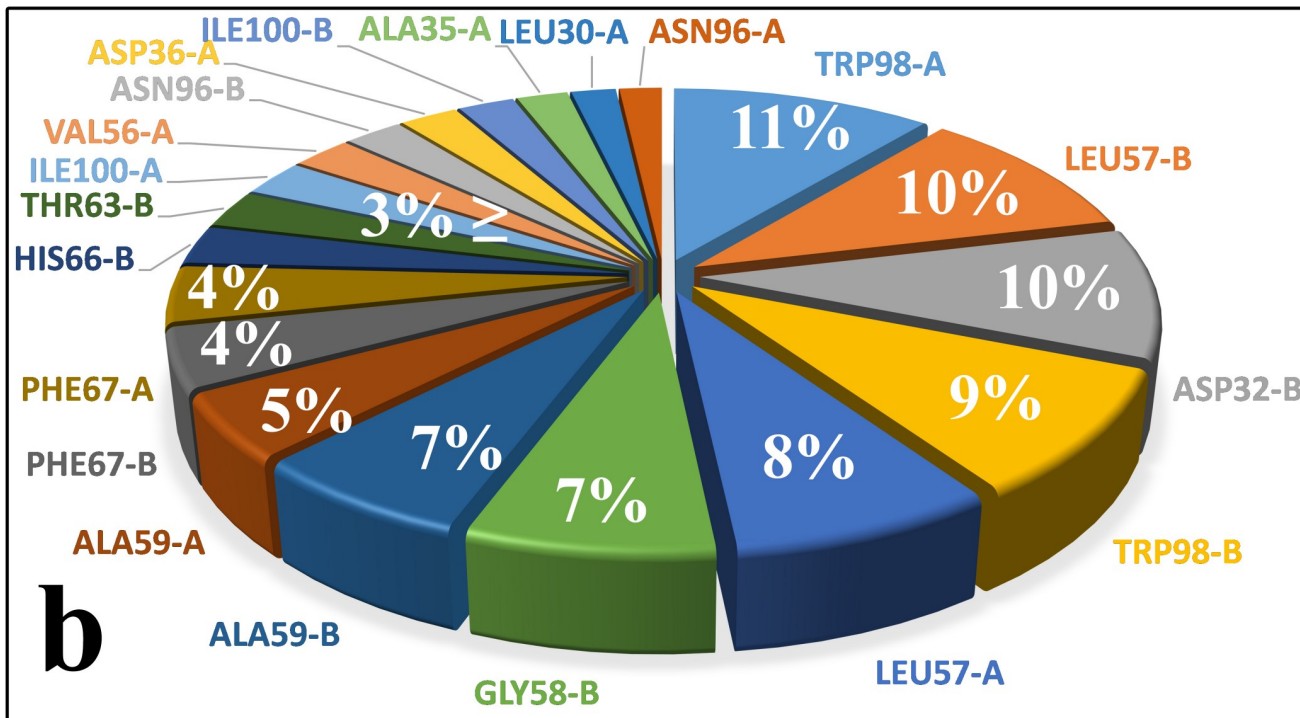

**Fig 5. The contribution of each residue from both A and B chains of the proteases in the total interaction energies between Indinavir and the enzyme in the unbinding pathways. a,** Indinavir-HIV protease complex, 1[st] replica. **b,** Indinavir-HTLV-1 protease complex, 2[nd] replica. A and B stands for "Chain A" and "Chain B", respectively. This data is the average interaction energy of each residue during the entire SuMD simulation.

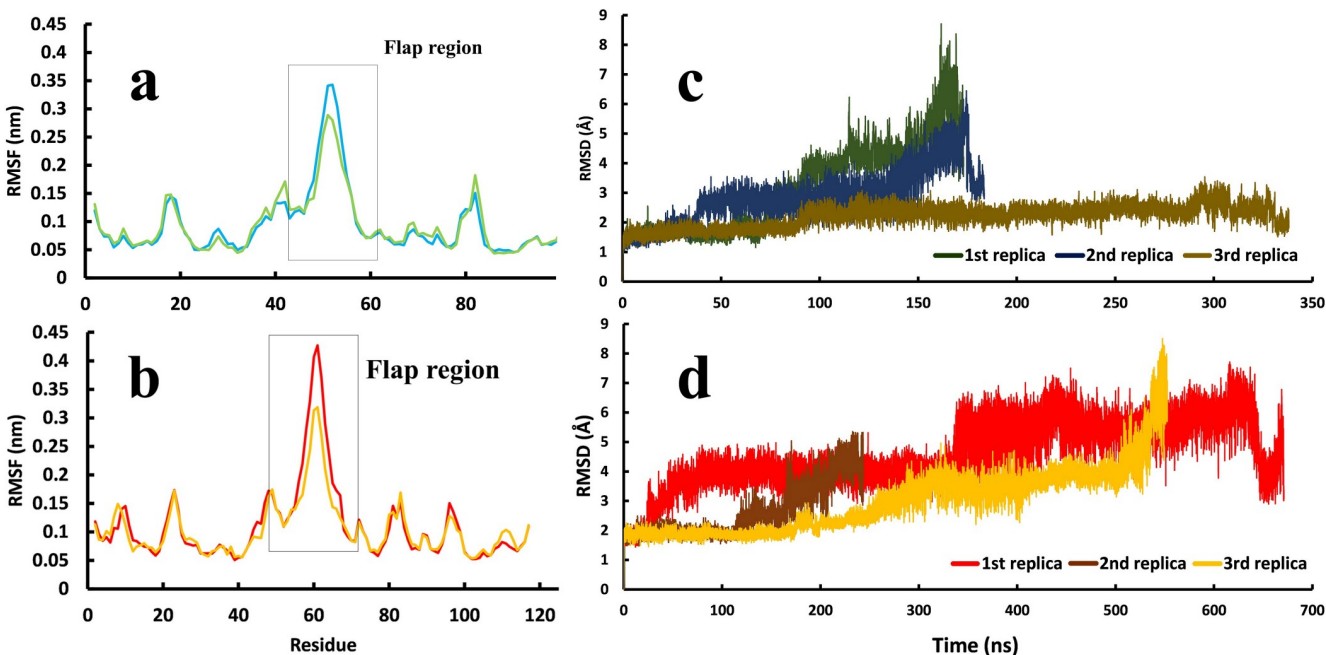

**Fig 6. The average RMSF values of residues and the RMSD values of the two proteases' backbone atoms in three replicas. a,** RMSF values of HIV protease and, **b,** RMSF values of HTLV-1 protease. **c,** RMSD values of the backbone atoms of HIV protease and, **d,** RMSD values of the backbone atoms of HTLV-1 protease.

with Indinavir and may explain another reason for the higher mobility of the flap region in HTLV-1 protease (Fig 6B) [72–74].

The flap region of the two proteases can acquire three states: closed, semi-open, and open [67,75]. In the 6 SuMD simulations performed in this study, three replicas for each complex, all of these states were observed in the unbinding pathways of Indinavir (Fig 7A and 7B). The natural function of proteases is to recognize the polypeptide chain, trap and place it in the active site where the catalytic residues cut the peptide bond at a specific position. This function of the flaps is highly crucial for the whole function of the enzyme [68,76]. The weak interaction between the ligand and the flap regions' residues coupled with the high mobility of this region causes the ligand to unbind and leave the enzymes. In almost all of the replicas, Indinavir left the active site from the flap region.

Indinavir interacts with these two enzymes at two separate states, the native and the intermediate states. In the native state, Asp32/Asp32′ in the HTLV-1 protease and Asp25/Asp25′ in HIV protease have the role of interacting with the ligand and make it stable. Although many other residues interact with the ligand in the native state through VdW and electrostatic interactions, the two Asp residues in the deep parts of both enzymes keep Indinavir anchored to the native state. One of the two catalytic Asp residues is always protonated, which is essential for the enzyme function [77–79]. This protonation enables one of the Asp residues to make a strong hydrogen bond with the central hydroxyl group of Indinavir. The first step of the unbinding pathway is the hydrogen bond breakage with the catalytic Asp residues in all replicas. The rotation of the hydroxyl group of Indinavir is the main cause of this breakage (Fig 8A and 8B). The rotation around $C_{10}$-$C_{11}$ bond and change of its dihedral angle at the start of the unbinding process (Fig 8D), and simultaneously the increasing distance between the $OD_2$ atom of Asp32 and the $H_{21}$ atom of Indinavir (Fig 8C) prove the breakage of this hydrogen bond and are the first events of the unbinding pathway (more examples in S4 Fig).

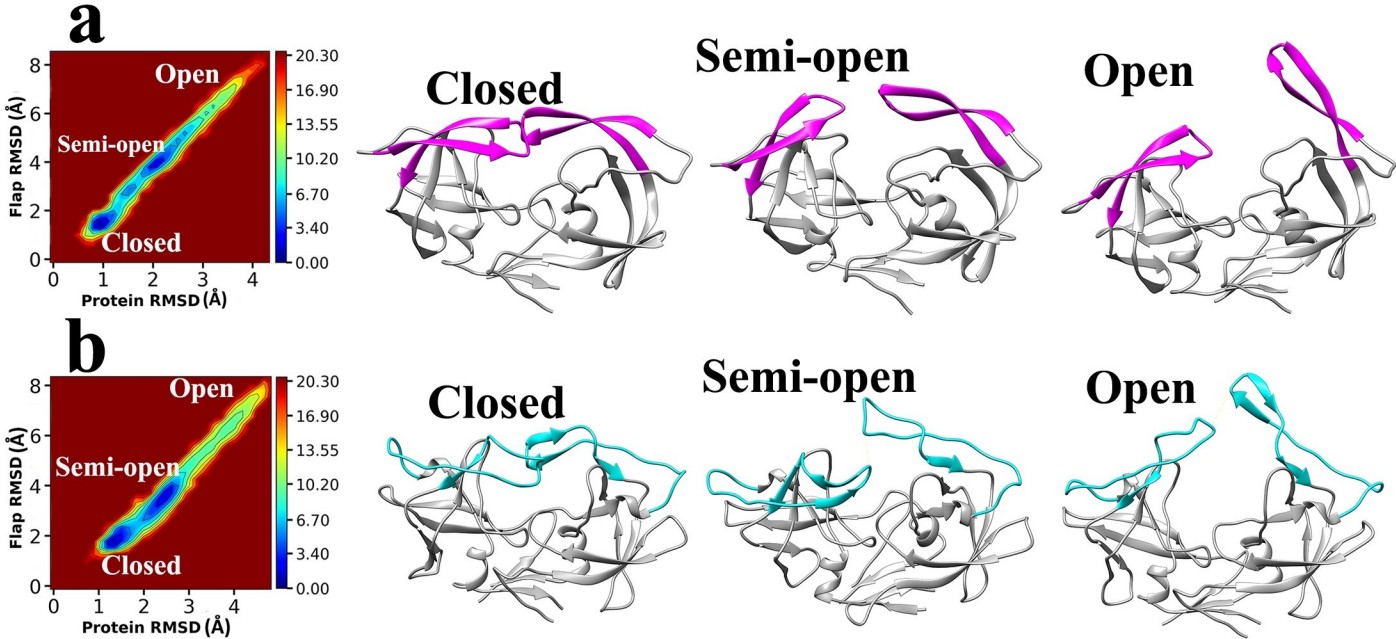

**Fig 7. The main conformational states of the flap region were acquired during the unbinding pathways of Indinavir. a,** The "out-of-equilibrium" free energy landscape and the 3D structures of HTLV-1 protease with different flap states. **b,** The "out-of-equilibrium" free energy landscape and the 3D structures of HIV protease with different flap states.

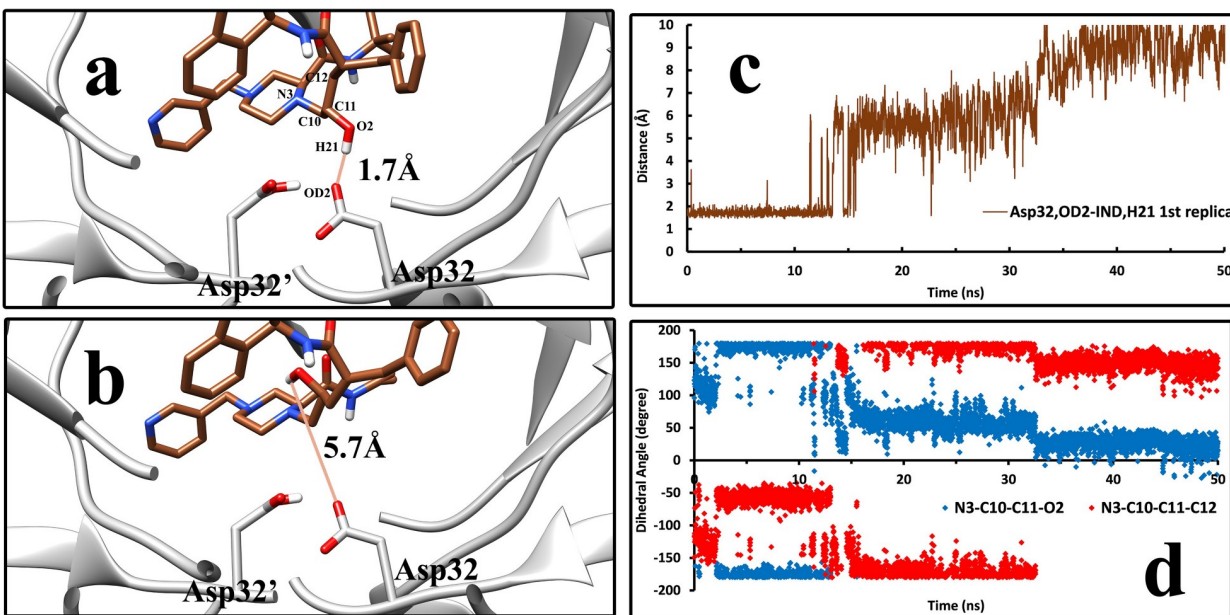

**Fig 8. The first step of the unbinding pathway of Indinavir. a,** Indinavir in the native state where a hydrogen bond with the Asp32 in the HTLV-1 protease keeps it stable in this state. **b,** the breakage of the hydrogen bond, which triggers the unbinding pathway. Both figures are frames 1153 and 1154, showing that this event needs very little time to happen. **c,** The distance between the $OD_2$ atom of Asp32 and the $H_{21}$ atom of Indinavir in the first 50 ns of the unbinding pathway of Indinavir-HTLV-1 complex in the 1st replica. **d,** The dihedral angles of the rotatable bond responsible for the rotation of the hydroxyl group of Indinavir in the first 50 ns of the unbinding pathway of Indinavir-HTLV-1 complex in the 1st replica.

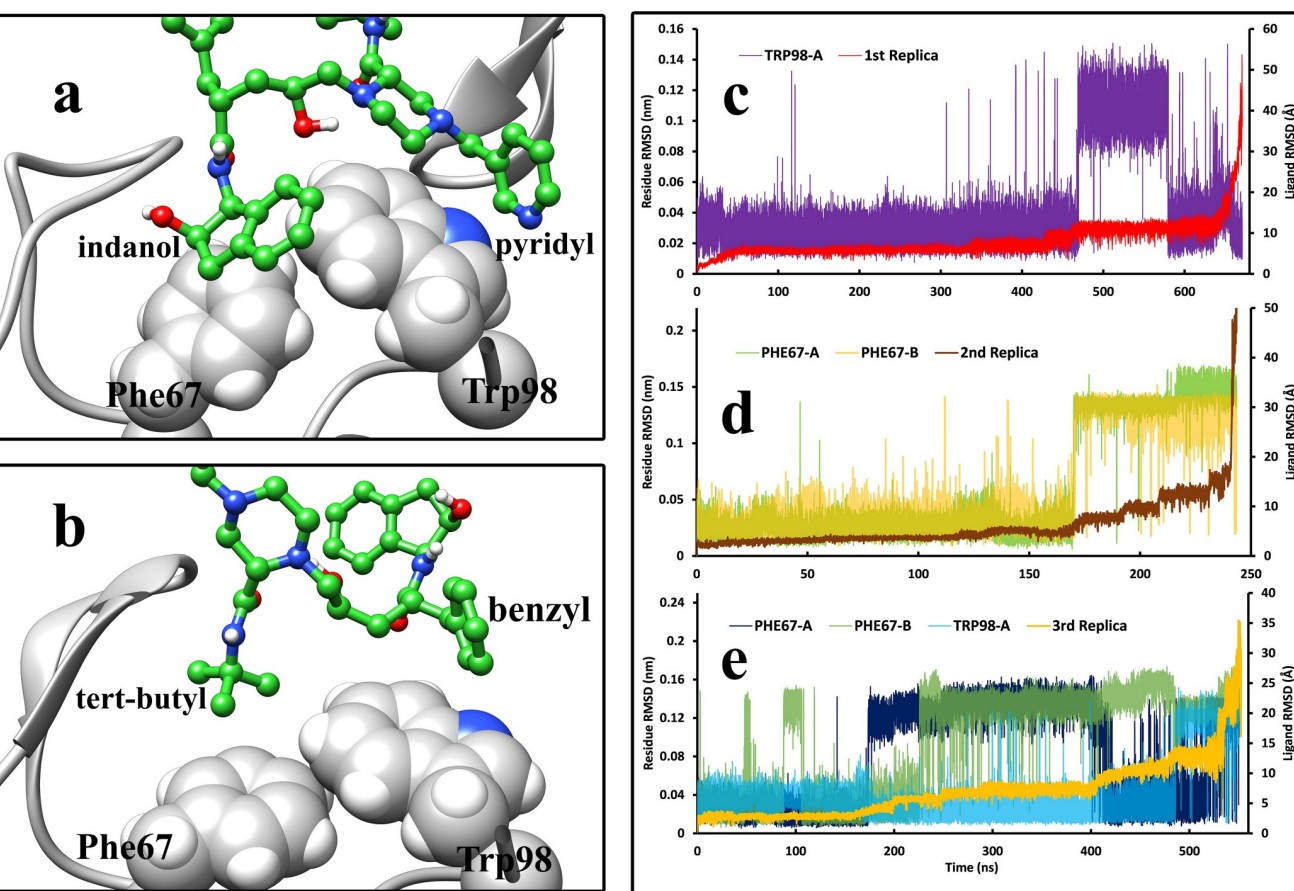

**Fig 9. The interactions of Indinavir moieties with the Phe 67 and Trp98 from both chains of HTLV-1 protease during the unbinding pathway and the comparison between the RMSD values of Indinavir (Å) and the RMSD values (nm) of critical aromatic residues such as Phe67 and Trp98 in both chains during the unbinding pathways in the three replicas. a**, indanol, pyridyl, and, **b**, tert-butyl and benzyl moieties of Indinavir interacting with the Indinavir molecule in different states. **c,** The first replica of the unbinding pathway where only Trp98 from chain A showed elevated RMSD values (nm) in comparison to Indinavir RMSD values (Å). **d,** The second replica where the two Phe67 from both chains presented their role in comparing the Indinavir RMSD values (Å) and the residues' RMSD values (nm). **e,** The third replica where Phe67 from both chains and Trp98 from chain A exhibited their role in the unbinding pathway.

After the first step, the unbinding pathway of Indinavir varies in some points between the HIV and HTLV-1 cases. After breaking the above-mentioned hydrogen bond, Indinavir started to get further from the active site's deep parts and interact with higher positioned residues and the flap region residues. One of the active site's critical points is the Trp98/Trp98′ residue in HTLV-1 protease, which Val82/Val82′ occupies in the HIV protease at the same position. The Trp residues in the HTLV-1 protease are one of the main contributors to the interaction energies. Phe67/Phe67′ was also an aromatic residue in the active site of HTLV-1 protease, which interacts with Indinavir in the native and intermediate states. These aromatic residues' interactions with the ligand make it easier for the ligand to get out of the native state (Fig 9C–9E). They mainly interact with the benzyl and the pyridyl moieties of Indinavir, but interactions with the indanol and tert-butyl- moieties were also spotted in the unbinding pathway (Figs 1B, 9A and 9B). It was observed that they could acquire different orientations and make edge-to-face and face-to-face π-stacking and π-alkyl interactions. After the intermediate states, the ligand in both cases interacted solely with the flap region's residues for a very short time and then left the enzymes ultimately and got solvated in the simulation box.

In 2003, Sa-Filho et al. provided very interesting mutagenesis data about the mutant forms of HIV protease in which V82F was one of the main contributors to drug resistance during treatment by Indinavir [80]. This mutation made Indinavir and other protease inhibitors act considerably less selectively. An even larger aromatic residue like Trp in this position in HTLV-1 protease can make a massive difference in the activity of the drug. This incredible experimental data guided us to explore the role of the aromatic residues inside the active site of HTLV-1 protease. The aromatic residues in the active site, Trp98, and Phe67 in both chains of this enzyme can make strong interactions with Indinavir in the intermediate state. These inter-actions are entropically unfavorable for the ligand since they make it easier for Indinavir to get out of the native binding mode and interact with higher positioned residues. By comparing the RMSD values of the ligand and the RMSD values of these essential residues of the HTLV-1 protease binding site during the unbinding pathway (Fig 9C–9E), the role of these aromatic residues, Phe67 and Trp98 were evident. In all three replicas of the Indinavir-HTLV-1 protease complex, the elevation of the RMSD values of these aromatic residues directly correlates to the elevated values of ligand RMSD. The transition of the ligand from the native state to the inter-mediate states directly corresponds to the conformational changes of these aromatic residues.

The fluctuation of every atom of the Indinavir molecule during the unbinding pathways in both enzymes (Fig 10A and 10B) shows that the fluctuation of the Indinavir atoms in the active site of HTLV-1 protease is different and slightly higher than that Indinavir in the active site of

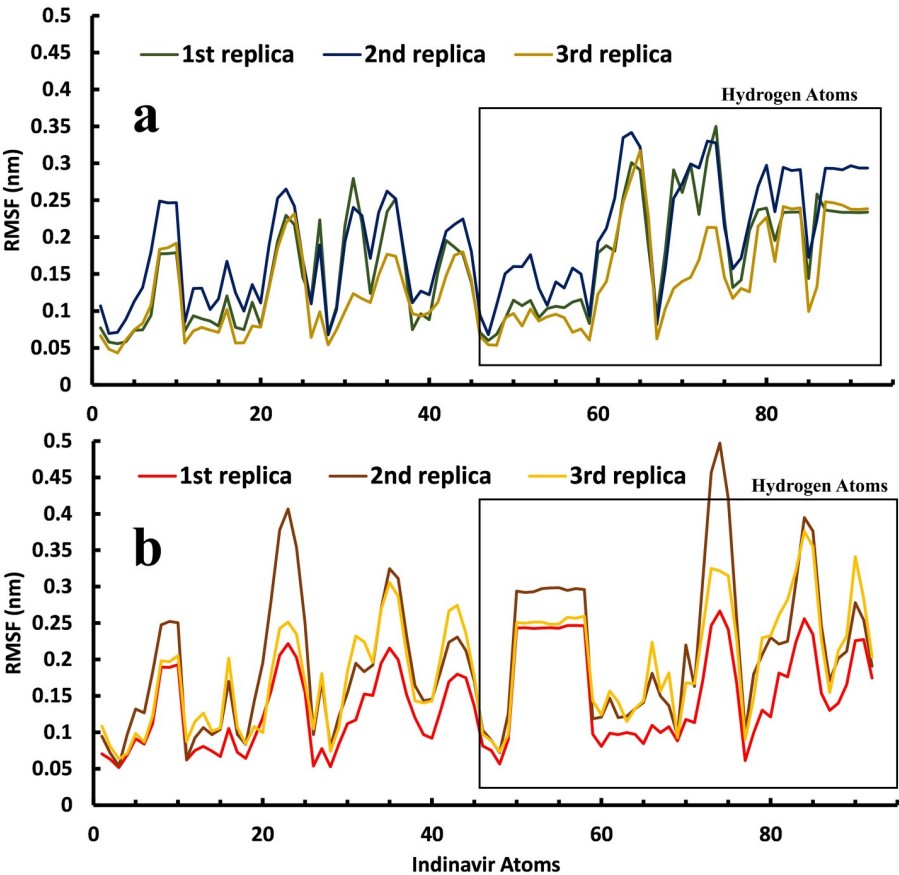

**Fig 10. A comparison between the RMSF values of the Indinavir atoms during the three replicas of the unbinding pathways. a,** in the active site of the HIV protease and, **b,** in the active site of the HTLV-1 protease.

the HIV protease. Consequently, the motions of Indinavir moieties in the active site of HTLV-1 protease are greater, and thus its interactions with the residues are less stable. This extra freedom in the active site of HTLV-1 protease makes it easier and more probable for Indinavir to interact with unfavorable regions of the active site, such as the flap region and the aromatic residues, discussed above, which facilitate the unbinding event. This data may explain another reason for the lack of inhibitory activity of this approved drug against the HTLV-1 protease.

As described in previous sections, the mobility and fluctuation of the flap region in the HTLV-1 protease is higher than HIV protease (Fig 6A and 6B). The flap region's residues in HIV protease have primarily organized secondary structures, two antiparallel β-strands (Fig 7B), whereas this region in the HTLV-1 protease is a mixture of short β-strands with random coils (Fig 7A). We believe that the higher mobility of the flap region coupled with the unfavorable interactions with aromatic residues in the active site of HTLV-1 protease can be the main reason for the significantly reduced inhibitory activity of Indinavir. A more effective inhibitor of HTLV-1 protease must make more interactions with the residues in the deep parts of the active site and have more stable interactions to reduce movements inside the active site. It must make more direct or water-mediated hydrogen bonds with the more inner positioned residues to avoid any interaction with the flap region since this region's high fluctuation makes it easy for the ligand to unbind. The aromatic residues are also a big part of the active site. For an inhibitor to stay in the active site, it must reduce the fluctuations of the aromatic residues and the fluctuations of the flap region residues. The results found in this work represent the dynamic details of the unbinding pathways of Indinavir from HTLV-1 and HIV proteases, and they are supporting the static results achieved by other research groups such as Kuhnert et al. [24]. We believe that all of the details achieved so far over the years can explain the lack of inhibitory activity of Indinavir or even any other approved protease inhibitor against HTLV-1 protease.

## Conclusion

Understanding the details of the unbinding pathway of potential small molecule inhibitors from their target proteins is of significant importance. One of the most promising tools for accomplishing this task is utilizing the SuMD simulation method. It is capable of reconstructing the binding and unbinding pathways of small molecule compounds. In this work, by taking advantage of this method, we reconstructed the unbinding pathway of Indinavir from the HTLV-1 protease and the HIV protease in order to understand the details of the unbinding pathways from both enzymes and also to compare the two unbinding events to find the reasons for the lack of inhibitory activity of Indinavir against the HTLV-1 protease. We achieved multiple unbinding pathways from both complexes and found that although the structure and the interaction profiles are broadly similar, superficial differences in the details cause massive effects on the inhibitor's performance. It was discovered that in the unbinding pathway of Indinavir form HTLV-1 protease, the fluctuation of the flap region is higher than that of HIV protease. We also found that aromatic residues in the active site of HTLV-1 protease, Phe67/Phe67′, and Trp98/Trp98′, are among the essential residues in the unbinding pathways and had the most contribution to the interaction energies. However, these interactions are unfavorable and cause the ligand to get out of the native binding mode. The unfavorable interactions coupled with the higher fluctuations of the flap region were the reasons for the poor inhibitory activity of Indinavir against HTLV-1 protease. We believe that the details found in this study can be an excellent assist in designing more effective compounds for inhibiting the HTLV-1 protease as a treatment solution for fighting the HTLV-1 virus.

## Supporting information

**S1 Fig. The distance between the COM of Indinavir and the COM of the Asp residues in the active site of HIV and HTLV-1 protease during the unbinding pathway.** a, b, c, The HIV protease, d, e, f, The HTLV-1 protease.
(TIF)

**S2 Fig. The VdW, electrostatic, and total interaction energies between Indinavir and the two proteases during the unbinding pathways. a, b, c,** The Indinavir-HIV complex and, **d, e, f**, The Indinavir- HTLV-1 protease complex.
(TIF)

**S3 Fig. The contribution of the active site residues to the total interaction energies in the unbinding pathways. a, b,** 1st, and 3rd replica of the Indinavir-HIV protease case. **c, b,** 1st, and 2nd replica of the Indinavir-HTLV-1 protease case.
(TIF)

**S4 Fig. The first step of the unbinding pathway of Indinavir is illustrated by (i) the distance of the hydrogen atom of the central hydroxyl group of Indinavir and the $OD_2$ atom of Asp32 in HTLV-1 protease and the Asp25 in HIV protease, and (ii) the dihedral angle of the $C_{10}$-$C_{11}$ bond in the Indinavir molecules. a**, The distance between the $OD_2$ atom of Asp25 and the $H_{21}$ atom of Indinavir in the first 50 ns of the unbinding pathway of Indinavir-HIV complex in the 2nd replica. **b**, The dihedral angles of the rotatable bond responsible for the rotation of the hydroxyl group of Indinavir in the first 50 ns of the unbinding pathway of the Indinavir-HIV complex in the 2nd replica. **c**, The distance between the $OD_2$ atom of Asp32 and the $H_{21}$ atom of Indinavir in the first 200 ns of the unbinding pathway of Indinavir-HTLV-1 complex in the 3rd replica. **d**, The dihedral angles of the rotatable bond responsible for the rotation of the hydroxyl group of Indinavir in the first 200 ns of the unbinding pathway of Indinavir-HTLV-1 complex in the 3rd replica.
(TIF)

**S1 File. HIV-rep1.mp4, the 1st unbinding pathway of Indinavir from the HIV protease enzyme.**
(MP4)

**S2 File. HIV-rep2.mp4, the 2nd unbinding pathway of Indinavir from the HIV protease enzyme.**
(MP4)

**S3 File. HIV-rep3.mp4, the 3rd unbinding pathway of Indinavir from the HIV protease enzyme.**
(MP4)

**S4 File. HTLV-1-rep1.mp4, the 1st unbinding pathway of Indinavir from the HTLV-1 protease enzyme.**
(MP4)

**S5 File. HTLV-1-rep2.mp4, the 2nd unbinding pathway of Indinavir from the HTLV-1 protease enzyme.**
(MP4)

**S6 File. HTLV-1-rep3.mp4, the 3rd unbinding pathway of Indinavir from the HTLV-1 protease enzyme.**
(MP4)

## Author Contributions

**Conceptualization:** Farzin Sohraby, Hassan Aryapour.

**Data curation:** Farzin Sohraby.

**Formal analysis:** Farzin Sohraby.

**Investigation:** Farzin Sohraby.

**Methodology:** Farzin Sohraby.

**Project administration:** Hassan Aryapour.

**Software:** Farzin Sohraby.

**Supervision:** Hassan Aryapour.

**Validation:** Farzin Sohraby.

**Visualization:** Farzin Sohraby.

**Writing – original draft:** Farzin Sohraby.

**Writing – review & editing:** Farzin Sohraby, Hassan Aryapour.

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
