## [Decision Letter · Decision Letter 0]

14 Jul 2021

PONE-D-21-18336

Comparative analysis of the unbinding pathways of antiviral drug Indinavir from HIV and HTLV1 proteases by Supervised Molecular Dynamics simulation

PLOS ONE

Dear Dr. Aryapour,

Thank you for submitting your manuscript to PLOS ONE. After careful consideration, we feel that it has merit but does not fully meet PLOS ONE’s publication criteria as it currently stands. Therefore, we invite you to submit a revised version of the manuscript that addresses the points raised during the review process.

In your revised manuscript please address as fully as possible the detailed and constructive comments of the two reviewers.

We look forward to receiving your revised manuscript.

Kind regards,

Israel Silman

Academic Editor

PLOS ONE

Journal Requirements:

"This investigation was supported by Golestan University, Gorgan, Iran."

"No, The author(s) received no specific funding for this work"

Reviewers' comments:

Reviewer's Responses to Questions

**Comments to the Author**

1. Is the manuscript technically sound, and do the data support the conclusions?

Reviewer #1: Yes

Reviewer #2: Partly

2. Has the statistical analysis been performed appropriately and rigorously? 

Reviewer #1: No

Reviewer #2: N/A

3. Have the authors made all data underlying the findings in their manuscript fully available?

Reviewer #1: Yes

Reviewer #2: No

4. Is the manuscript presented in an intelligible fashion and written in standard English?

Reviewer #1: Yes

Reviewer #2: Yes

5. Review Comments to the Author

Reviewer #1: This article is a theoretical study of the ligand unbinding of HIV and HTLV-1 proteases. Through the supervised molecular dynamics (SuMD) simulations on the two proteases in complex with the indinavir inhibitor, the authors observed multiple unbinding processes. Subsequently, the detailed analyses of these unbinding pathways suggested that the unfavorable interactions from Phe67/Phe67′, and Trp98/Trp98′ to the inhibitor, coupled with the higher fluctuations of the flap region of HTLV-1 protease are the reasons for the poor inhibitory activity of indinavir against HTLV-1 as compared to that against HIV protease. HIV and HTLV-1 proteases (particularly the former) are two extremely popular biomolecule systems for MD simulations to study the structural dynamics, protease-inhibitor interactions, et al. The observations in the present study are, to be honest, not novel as compared to the previous various simulation studies. The main issues to be addressed include:

1) All the conclusions are made based on three observed unbinding events per protease. The simulation data seems not solid enough.

2) SuMD yielded the time of 173, 183.5, and 338 ns required for the unbinding of the indinavir from HIV protease. Although no biased force is added in the simulation, SuMD is certainly different to the conventional MD and the simulated unbinding times are not the times in the reality. Then the direct comparison of the unbinding times between HIV and HTLV-1 proteases should be meaningless. The authors need to discuss it in the manuscript.

3) I couldn’t understand how the conclusions about the more flexible and fluctuated flaps of HTLV-1 than that of HIV were made from Fig. 6 and higher fluctuation of the indinavir atoms in the active site of HTLV-1 than in HIV were made from Fig. 10. The authors need to somehow redraw these figures to let people clearly know how these conclusions are drawn.

4) The authors mentioned that the simulation results in this work can help design more effective and more selective inhibitors for the HTLV-1 protease. Unfortunately, detailed discussion is absent in the article.

Reviewer #2: Obtaining and understanding the ligand unbinding mechanism and pathways is an important task in computational aided drug discovery. In this paper, the author use an unbiased, but out-of-equilibrium, MD simulation method, called supervised unbiased MD (SuMD), and apply it to a challenging systems. The topic of ligand unbinding is of general interested and should be of interest to the readership of PLoS One.

However, in the correct form, I am of the opinion that the manuscript is not of sufficient quality to be accepted for publication in PloS One (or other journals). The manuscript has to be significantly improved to be accepted for publication. Once the authors have made substantial changes to revise the manuscript, I am ready to reconsider my assessment.

I note that I reviewed a different manuscript of the same authors for Scientific Reports a few months ago. In that manuscript, the author also used SuMD and applied to a different systems. Some of my comments below regarding the SuMD method and the general presentation of results are nearly the same as I had for that manuscript. Therefore, the authors should have been aware of these issues.

In following, I list my comments and suggestions to improve the manuscript.

* A major issue that I have with the manuscript is that the authors heavily emphasize that SuMD is an unbiased method, while not fully acknowledging that the method is out-of-equilibrium due to the iterative selection of starting points for the subsequent simulations. Thus, the methods cannot give reliable kinetics (this is acknowledged for SuMD in Ref 41 of the manuscript). Furthermore, this makes SuMD rather similar to other out-of-equilibrium methods that use unbiased simulations like weighted ensemble, forward flux sampling, milestoning, and more (some of which allow for obtaining kinetics under certain conditions). This point should be discussed in manuscript and appropriate citations added if they not already included.

* Lines 85-87: "Unraveling the drug′s unbinding pathway from its target protein is an excellent task that can be very useful to understand the process′s mechanism and the residues involved" This sentence does not make sense, I ask the authors to re-phrase it.

* Lines 88-89: "One of the most accurate approaches for this task is the Supervised MD (SuMD) simulations". To describe SuMD as "One of the most accurate approaches" is a very strong statement by the authors. Do they have results that can be used to back-up this statement? For example, work comparing SuMD to other methods. If not, I would ask them to tone down this statement.

* Line 93-94: "This atomistic approach [SuMD] is entirely unbiased and is very accurate." Following the previous comment. To say that SuMD is "very accurate" is a rather strong statement by the authors. Do they have results that can be used to back-up this statement? For example, work comparing SuMD with other methods. If not, I would ask them to tone down this statement. Furthermore, as SuMD cannot give the correct kinetics or binding energies (e.g., discussed in Ref 41), it would perhaps not correct to call it "very accurate".

* Line 93-94: "This atomistic approach [SuMD] is entirely unbiased and is very accurate." While it is true that SuMD use unbiased simulations, it is not an equilibrium method. Rather I would characterize it as an out-of-equilibrium method. For this reason, while the unbinding pathways obtained from SuMD might be correct, SuMD cannot give the correct kinetics or binding energies. This is for example discussed in Ref 41. I would ask the authors to acknowledge that SuMD is not an equilibrium method.

* Line 93-94: Following my previous comment, I find a similarity of SuMD (and HSuMD) to other out-of-equilbrium methods like weighted ensemble, forward flux sampling, milestoning, and others. Many of these methods have been applied to ligand unbinding. Some of these methods can, under certain conditions, also obtain the correct kinetic parameters. I ask that the author to add citations to other similar out-of-equilibrium methods.

* Line 40: "HIV has been the most advertised virus on planet earth." I don't think that this is right phrase to use here.

* I assume that there have been many other papers that have investigated the HTLV-1 and HIV protease using different molecular simulation techniques. However, I find missing in the introduction proper discussion about these other paper. This needs to be added.

* Line 108-109: The MD simulations time step is not listed as it should be. This should be added.

* Line 112: The number of water molecules included in the two systems is missing. This needs to be added.

* Line 119-121: "The modified Berendsen (V-rescale) thermostat [55] and Parrinello–Rahman barostat [56] respectively were applied for 100 and 300 ps to keep the system in the stable environmental conditions (310 K, 1 Bar)." This is a bit weird statement, it sounds like a thermostat or a barostat was not used for the production runs. I assume what the authors mean that they did 100 ps of NVT equilibration and 300 ps of NPT equilibration. This should be clarified.

* Line 121-122: Given the previous comment, were the production runs done in NVT or NPT conditions? If so, what were the thermo- and barostats used (I assume the same as for the equilibration?) This should be clarified. Furthermore, parameters for the thermo- and barostat should be listed, like relaxation times, etc.

* Line 121-122: "Finally, SuMD simulations [57] were carried out under the periodic boundary conditions (PBC), set at XYZ coordinates to ensure that the atoms had stayed inside the simulation box," This is also a strange statement, it sound like PBC were not used for the previous energy minimization and the equilibration runs (given that the authors used Gromacs, PBC should have been used there). Therefore, this should be clarified. Furthermore, the phrase "set at XYZ coordinates to ensure that the atoms had stayed inside the simulation box," is very unclear, what do the authors mean here? That they removed center of mass motion? This needs to be clarified.

* Line 125-126: "The free energy landscapes were rendered using Matplotlib [60]." As I mention below, it would be more correct to call these "out-of-equilibrium free energy landscapes"

* Line 126-127: "In addition, to estimate the binding free energy we used the g_mmpbsa package [61]." In the manuscript, the authors never mention binding free energies again, so I assume this text should not be there, maybe it was not removed by mistake. The authors need to fix this.

* Line 129-130: "the entire simulation is divided into a series of replicas, and a specific parameter is monitored throughout them as the guideline to choose the starting point of the next replica." How was the SuMD procedure implemented? Is it done in an external script and using some other codes? (For example in Ref 41, PLUMED was used to calculate distances.) This should be detailed in the manuscript.

* Lines 132-135: "For example, in our series of replicas, we considered the distance between the Center Of Mass (COM) of the drug and the COM of the Asp32 and Asp32′ in the HTLV-1 and the Asp25 and the Asp25′ in the HIV proteases as the guideline for selecting the best frame to be the starting point of the next replica." For full reproducibility, more details should be listed for how the distance in the SuMD simulations is defined. For the ligand, are all atoms taking into account or only heavy atoms? The same for the amino acid residues, it should be listed in the what atoms on the residues are considered (perhaps in the SI).

* Lines 132-135: "For example, in our series of replicas," I don't see why "For example" is needed in this sentence.

* Lines 132-135: "the COM of the Asp32 and Asp32′ in the HTLV-1 and the Asp25 and the Asp25′ in the HIV proteases" This is unclear to me. Does this mean that the author calculated two distances for each system? If so, how was this used in the SuMD procedure? Did they use the maximum of the two distances? This should be clarified.

* Lines 133: "Asp32 and Asp32'" I am not an expert on the specific system so it is unclear to me what is the difference between "Asp32" and "Asp32'". From Figure 5, I would assume that "Asp32" means amino acid from chain A and "Asp32'" means amino acid from chain B. The authors cannot expect the reader to know and understand this notion used for the amino acids. Therefore, I ask that the authors explicitly define the notation used for the amino acids and what is the difference between "Asp32" and "Asp32'".

* Lines 176-178: "There are no biasing forces involved in the simulations, and they are entirely unbiased. The only difference between this method and conventional MD simulations is the automatic supervision at choosing the most appropriate frame in a replica for extending the simulation." This discussion ignores the fact that the frame selection creates an out-of equilibrium effect. Therefore, even though SuMD is an unbiased method, it is an out-of-equilibrium method. This should be acknowledged here.

* Figure 2: How is the RMSD of the Indinavir defined? This should be detailed in the caption of Figure 2.

* Figure 2: For the six simulations considered, I would like to see in the results, or in the SI, the time series of the distances that are used in the SuMD. This is useful for reader to understand better how the SuMD methods acts for these systems.

* Line 190-191: "In total, three stable states were observed during the unbinding pathways; (i) the Native state (N), (ii) the Intermediate states (I1, I2) and, (iii) the Solvated state (S) (Fig. 3a-f)." The Figure 3 that the authors refer to includes what the authors call "Free energy landscapes (FEL) of the unbinding pathways". I agree that showing the unbinding obtained from SuMD in this manner is useful. However, these are not "free energy landscapes" in traditional meaning of the notation as SuMD is an out-of-equilibrium method. Thus, I don't think it is correct to call this a FEL, at least in equilibrium meaning of a FEL. I would ask the authors to make it clear in the text that the the surfaces in fig. 3a-f are not traditional equilibrium free energy landscapes, but rather out-of-equilibrium.

* Figure 3, line 204: "Free energy landscapes (FEL) of the unbinding pathways" same as previous comment. I ask the authors to make it clear that the FEL shown in Figure 3 are not traditional equilibrium free energy landscapes, but rather out-of-equilibrium.

* Figure 3: For the reader to be able to compare the different surfaces better, I would suggest the authors to use the same color scale in all figures. Panel (d) has a different color scale (from 0 to 23.70) than all other panels (from 0 to 20.30).

* Figure 5: How are the contribution of the different residues to the interaction energy obtained? Are they averaged over some part of the simulation? This should be detailed in the caption of Figure 5.

* Figure 5: In Figure 5, the authors talk about chain A and B and use the notation Ala59-A and Ala59-B (for example). However, elsewhere in the manuscript the authors use the notation Ala59 and Ala59' (for example) that I assume mean that Ala59 would be from chain A and Ala59' is from chain B. To avoid confusion, the authors should use the same notation throughout the manuscript. Therefore, this should be changed either in Figure 5 or everywhere else in the manuscript.

* Figure 6: For the reader to be able to compare the different systems better I would ask the authors to use the same y-scale in panels (a) and (b), so that both go from 0 to 0.45.

* Figure 6: For the reader to be able to compare the different systems better I would ask the authors to use the same y-scale in panels (c) and (d), so that both go from 0 to 0.9.

* Figure 7: lines 285 and 286: "The free energy landscape" Same as for Figure 3, I ask the authors to make it clear that the FEL shown in Figure 7 are not traditional equilibrium free energy landscapes, but rather out-of-equilibrium.

* Figure 7: Same as for Figure 3, for the reader to be able to compare the different surfaces better, I would suggest the authors to use the same color scale in panels (a) and (b).

* Regarding Data Availability Statement. The authors state "All data will be available online" without stating how it will be made available. I don't think this is line with PLoS One policy and needs to be fixed before submission. The authors could for example upload the data to Zenodo.

6. PLOS authors have the option to publish the peer review history of their article (what does this mean?). If published, this will include your full peer review and any attached files.

Reviewer #1: No

Reviewer #2: No

---

## [Author Response · Author response to Decision Letter 0]

23 Jul 2021

Dear Editor

Thanks for the constructive comments on the manuscript. We are pleased to submit the revised version of our original research paper entitled “Comparative analysis of the unbinding pathways of antiviral drug Indinavir from HIV and HTLV1 proteases by Supervised Molecular Dynamics simulation” for publication in your journal. We carefully applied the reviewers’ comments in the text and the figures of the manuscript and hope that this version is appropriate for publication. We would be honored to be a part of your impressive journal.

Funding: This study was supported by Golestan University, Gorgan, Iran.

Thank you in advance for your consideration

Yours sincerely,

Hassan Aryapour

Reviewer #1: 

This article is a theoretical study of the ligand unbinding of HIV and HTLV-1 proteases. Through the supervised molecular dynamics (SuMD) simulations on the two proteases in complex with the indinavir inhibitor, the authors observed multiple unbinding processes. Subsequently, the detailed analyses of these unbinding pathways suggested that the unfavorable interactions from Phe67/Phe67′, and Trp98/Trp98′ to the inhibitor, coupled with the higher fluctuations of the flap region of HTLV-1 protease are the reasons for the poor inhibitory activity of indinavir against HTLV-1 as compared to that against HIV protease. HIV and HTLV-1 proteases (particularly the former) are two extremely popular biomolecule systems for MD simulations to study the structural dynamics, protease-inhibitor interactions, et al. The observations in the present study are, to be honest, not novel as compared to the previous various simulation studies. The main issues to be addressed include:

Question #1: All the conclusions are made based on three observed unbinding events per protease. The simulation data seems not solid enough.

Answer #1: Three SuMD production runs have been performed for each enzyme and the data achieved are all consistent. We believe reconstructing three unbinding events was adequate to reach a scientific conclusion. However, each ligand may have an infinite number of unbinding paths, and it is impossible to reach all of them. Nevertheless, the more replicas there are, the results will be closer to reality.

Question #2: SuMD yielded the time of 173, 183.5, and 338 ns required for the unbinding of the indinavir from HIV protease. Although no biased force is added in the simulation, SuMD is certainly different from the conventional MD and the simulated unbinding times are not the times in reality. Then the direct comparison of the unbinding times between HIV and HTLV-1 proteases should be meaningless. The authors need to discuss it in the manuscript.

Answer #2: An explanation describing this point is present in the text: “Although Indinavir inhibits HIV protease much more selectively than the HTLV-1 protease, the overall time needed for the unbinding events of HTLV-1 protease case was considerably more than that of the HIV protease case. However, many factors govern an inhibitor's activity, and only comparing the overall duration of the unbinding events in three series of replicas is not a correct way of comparing these two cases”.

Question #3: I couldn’t understand how the conclusions about the more flexible and fluctuated flaps of HTLV-1 than that of HIV were made from Fig. 6 and higher fluctuation of the indinavir atoms in the active site of HTLV-1 than in HIV were made from Fig. 10. The authors need to somehow redraw these figures to let people clearly know how these conclusions are drawn.

Answer #3: In Fig. 6, the RMSF values of the two proteases have been shown in which the values of the flap region of the HTLV-1 protease are noticeably higher than that of HIV protease. Also, in Fig. 10, the RMSF values of the Indinavir atoms have been compared in which these values are higher as well. We think the discussion in the text of the manuscript and the caption are self-explanatory. 

Question #4: The authors mentioned that the simulation results in this work can help design more effective and more selective inhibitors for the HTLV-1 protease. Unfortunately, detailed discussion is absent in the article.

Answer #4: The last paragraph of the discussion section describes some suggestions for designing a more effective inhibitor for the HTLV-1 protease.

Reviewer #2: Obtaining and understanding the ligand unbinding mechanism and pathways is an important task in computational aided drug discovery. In this paper, the author use an unbiased, but out-of-equilibrium, MD simulation method, called supervised unbiased MD (SuMD), and apply it to a challenging systems. The topic of ligand unbinding is of general interested and should be of interest to the readership of PLoS One.

However, in the correct form, I am of the opinion that the manuscript is not of sufficient quality to be accepted for publication in PloS One (or other journals). The manuscript has to be significantly improved to be accepted for publication. Once the authors have made substantial changes to revise the manuscript, I am ready to reconsider my assessment.

I note that I reviewed a different manuscript of the same authors for Scientific Reports a few months ago. In that manuscript, the author also used SuMD and applied to a different systems. Some of my comments below regarding the SuMD method and the general presentation of results are nearly the same as I had for that manuscript. Therefore, the authors should have been aware of these issues.

In following, I list my comments and suggestions to improve the manuscript.

Question #1: A major issue that I have with the manuscript is that the authors heavily emphasize that SuMD is an unbiased method, while not fully acknowledging that the method is out-of-equilibrium due to the iterative selection of starting points for the subsequent simulations. Thus, the methods cannot give reliable kinetics (this is acknowledged for SuMD in Ref 41 of the manuscript). Furthermore, this makes SuMD rather similar to other out-of-equilibrium methods that use unbiased simulations like weighted ensemble, forward flux sampling, milestoning, and more (some of which allow for obtaining kinetics under certain conditions). This point should be discussed in manuscript and appropriate citations added if they not already included.

Answer #1: The SuMD method counts as an unbiased method, and it has been used by several other scientific groups which are cited in the text. However, in the methodology, we addressed the out-of-equilibration problem by running NVT, and NPT equilibrium runs after every frame selection before the 500 ps production runs. Therefore, the algorithm we have written to perform the SuMD is NOT out-of-equilibrium. An explanation about this has been added to the text.

Question #2: Lines 85-87: "Unraveling the drug′s unbinding pathway from its target protein is an excellent task that can be very useful to understand the process′s mechanism and the residues involved" This sentence does not make sense, I ask the authors to re-phrase it.

Answer #2: The statement has been Improved.

Question #3: Lines 88-89: "One of the most accurate approaches for this task is the Supervised MD (SuMD) simulations". To describe SuMD as "One of the most accurate approaches" is a very strong statement by the authors. Do they have results that can be used to back-up this statement? For example, work comparing SuMD to other methods. If not, I would ask them to tone down this statement.

Answer #3: The statement has been Improved.

Question #4: Line 93-94: "This atomistic approach [SuMD] is entirely unbiased and is very accurate." Following the previous comment. To say that SuMD is "very accurate" is a rather strong statement by the authors. Do they have results that can be used to back-up this statement? For example, work comparing SuMD with other methods. If not, I would ask them to tone down this statement. Furthermore, as SuMD cannot give the correct kinetics or binding energies (e.g., discussed in Ref 41), it would perhaps not correct to call it "very accurate".

Answer #4: The statement has been Improved.

Question #5: Line 93-94: "This atomistic approach [SuMD] is entirely unbiased and is very accurate." While it is true that SuMD use unbiased simulations, it is not an equilibrium method. Rather I would characterize it as an out-of-equilibrium method. For this reason, while the unbinding pathways obtained from SuMD might be correct, SuMD cannot give the correct kinetics or binding energies. This is for example discussed in Ref 41. I would ask the authors to acknowledge that SuMD is not an equilibrium method.

Answer #5: An explanation was added.

Question #6: Line 93-94: Following my previous comment, I find a similarity of SuMD (and HSuMD) to other out-of-equilbrium methods like , milestoning, and others. Many of these methods have been applied to ligand unbinding. Some of these methods can, under certain conditions, also obtain the correct kinetic parameters. I ask that the author to add citations to other similar out-of-equilibrium methods.

Answer #6: citations to other similar methods were added.

Question #7: Line 40: "HIV has been the most advertised virus on planet earth." I don't think that this is right phrase to use here.

Answer #7: The statement has been Improved.

Question #8: I assume that there have been many other papers that have investigated the HTLV-1 and HIV protease using different molecular simulation techniques. However, I find missing in the introduction proper discussion about these other paper. This needs to be added.

Answer #8: There are numerous papers about the MD simulation HIV protease but very few about the MD simulation of HTLV-1 protease. Worse still, nearly no paper exists that present and compare the dynamical conformational details of these two important enzymes.

Question #9: Line 108-109: The MD simulations time step is not listed as it should be. This should be added.

Answer #9: added.

Question #10: Line 112: The number of water molecules included in the two systems is missing. This needs to be added.

Answer #10: added.

Question #11: Line 119-121: "The modified Berendsen (V-rescale) thermostat [55] and Parrinello–Rahman barostat [56] respectively were applied for 100 and 300 ps to keep the system in the stable environmental conditions (310 K, 1 Bar)." This is a bit weird statement, it sounds like a thermostat or a barostat was not used for the production runs. I assume what the authors mean that they did 100 ps of NVT equilibration and 300 ps of NPT equilibration. This should be clarified.

Answer #11: The text has been Improved.

Question #12: Line 121-122: Given the previous comment, were the production runs done in NVT or NPT conditions? If so, what were the thermo- and barostats used (I assume the same as for the equilibration?) This should be clarified. Furthermore, parameters for the thermo- and barostat should be listed, like relaxation times, etc.

Answer #12: The text has been Improved and clarified. 

Question #13: Line 121-122: "Finally, SuMD simulations [57] were carried out under the periodic boundary conditions (PBC), set at XYZ coordinates to ensure that the atoms had stayed inside the simulation box," This is also a strange statement, it sound like PBC were not used for the previous energy minimization and the equilibration runs (given that the authors used Gromacs, PBC should have been used there). Therefore, this should be clarified. Furthermore, the phrase "set at XYZ coordinates to ensure that the atoms had stayed inside the simulation box," is very unclear, what do the authors mean here? That they removed center of mass motion? This needs to be clarified.

Answer #13: The text has been Improved and clarified. 

Question #14: Line 125-126: "The free energy landscapes were rendered using Matplotlib [60]." As I mention below, it would be more correct to call these "out-of-equilibrium free energy landscapes"

Answer #14: refer to the Answer #1.

Question #15: Line 126-127: "In addition, to estimate the binding free energy we used the g_mmpbsa package [61]." In the manuscript, the authors never mention binding free energies again, so I assume this text should not be there, maybe it was not removed by mistake. The authors need to fix this.

Answer #15: The text was fixed. We calculated the interaction energies by g-mmpbsa. 

Question #16: Line 129-130: "the entire simulation is divided into a series of replicas, and a specific parameter is monitored throughout them as the guideline to choose the starting point of the next replica." How was the SuMD procedure implemented? Is it done in an external script and using some other codes? (For example in Ref 41, PLUMED was used to calculate distances.) This should be detailed in the manuscript.

Answer #16: An explanation was added.

Question #17: Lines 132-135: "For example, in our series of replicas, we considered the distance between the Center Of Mass (COM) of the drug and the COM of the Asp32 and Asp32′ in the HTLV-1 and the Asp25 and the Asp25′ in the HIV proteases as the guideline for selecting the best frame to be the starting point of the next replica." For full reproducibility, more details should be listed for how the distance in the SuMD simulations is defined. For the ligand, are all atoms taking into account or only heavy atoms? The same for the amino acid residues, it should be listed in the what atoms on the residues are considered (perhaps in the SI).

Answer #17: corrected.

Question #18: Lines 132-135: "For example, in our series of replicas," I don't see why "For example" is needed in this sentence.

Answer #18: corrected.

Question #19: Lines 132-135: "the COM of the Asp32 and Asp32′ in the HTLV-1 and the Asp25 and the Asp25′ in the HIV proteases" This is unclear to me. Does this mean that the author calculated two distances for each system? If so, how was this used in the SuMD procedure? Did they use the maximum of the two distances? This should be clarified.

Answer #19: Corrected.

Question #20: Lines 133: "Asp32 and Asp32'" I am not an expert on the specific system so it is unclear to me what is the difference between "Asp32" and "Asp32'". From Figure 5, I would assume that "Asp32" means amino acid from chain A and "Asp32'" means amino acid from chain B. The authors cannot expect the reader to know and understand this notion used for the amino acids. Therefore, I ask that the authors explicitly define the notation used for the amino acids and what is the difference between "Asp32" and "Asp32'".

Answer #20: Corrected.

Question #21: Lines 176-178: "There are no biasing forces involved in the simulations, and they are entirely unbiased. The only difference between this method and conventional MD simulations is the automatic supervision at choosing the most appropriate frame in a replica for extending the simulation." This discussion ignores the fact that the frame selection creates an out-of equilibrium effect. Therefore, even though SuMD is an unbiased method, it is an out-of-equilibrium method. This should be acknowledged here.

Answer #21: refer to the Answer #1.

Question #22: Figure 2: How is the RMSD of the Indinavir defined? This should be detailed in the caption of Figure 2.

Answer #22: Corrected.

Question #23: Figure 2: For the six simulations considered, I would like to see in the results, or in the SI, the time series of the distances that are used in the SuMD. This is useful for reader to understand better how the SuMD methods acts for these systems.

Answer #23: Added to the SI.

Question #24: Line 190-191: "In total, three stable states were observed during the unbinding pathways; (i) the Native state (N), (ii) the Intermediate states (I1, I2) and, (iii) the Solvated state (S) (Fig. 3a-f)." The Figure 3 that the authors refer to includes what the authors call "Free energy landscapes (FEL) of the unbinding pathways". I agree that showing the unbinding obtained from SuMD in this manner is useful. However, these are not "free energy landscapes" in traditional meaning of the notation as SuMD is an out-of-equilibrium method. Thus, I don't think it is correct to call this a FEL, at least in equilibrium meaning of a FEL. I would ask the authors to make it clear in the text that the the surfaces in fig. 3a-f are not traditional equilibrium free energy landscapes, but rather out-of-equilibrium.

Answer #24: refer to the Answer #1.

Question #25: Figure 3, line 204: "Free energy landscapes (FEL) of the unbinding pathways" same as previous comment. I ask the authors to make it clear that the FEL shown in Figure 3 are not traditional equilibrium free energy landscapes, but rather out-of-equilibrium.

Answer #25: refer to the Answer #1.

Question #26: Figure 3: For the reader to be able to compare the different surfaces better, I would suggest the authors to use the same color scale in all figures. Panel (d) has a different color scale (from 0 to 23.70) than all other panels (from 0 to 20.30).

Answer #26: corrected

Question #27: Figure 5: How are the contribution of the different residues to the interaction energy obtained? Are they averaged over some part of the simulation? This should be detailed in the caption of Figure 5.

Answer #27: corrected

Question #28: Figure 5: In Figure 5, the authors talk about chain A and B and use the notation Ala59-A and Ala59-B (for example). However, elsewhere in the manuscript the authors use the notation Ala59 and Ala59' (for example) that I assume mean that Ala59 would be from chain A and Ala59' is from chain B. To avoid confusion, the authors should use the same notation throughout the manuscript. Therefore, this should be changed either in Figure 5 or everywhere else in the manuscript.

Answer #28: Both forms have been used for the text and also for the figures and we have made them clear throughout the manuscript. We believe this does not make any complications for the readers.

Question #29: Figure 6: For the reader to be able to compare the different systems better I would ask the authors to use the same y-scale in panels (a) and (b), so that both go from 0 to 0.45.

Answer #29: Corrected

Question #30: Figure 6: For the reader to be able to compare the different systems better I would ask the authors to use the same y-scale in panels (c) and (d), so that both go from 0 to 0.9.

Answer #30: Corrected

Question #31: Figure 7: lines 285 and 286: "The free energy landscape" Same as for Figure 3, I ask the authors to make it clear that the FEL shown in Figure 7 are not traditional equilibrium free energy landscapes, but rather out-of-equilibrium.

Answer #31: refer to the Answer #1.

Question #32: Figure 7: Same as for Figure 3, for the reader to be able to compare the different surfaces better, I would suggest the authors to use the same color scale in panels (a) and (b).

Answer #32: Corrected

---

## [Decision Letter · Decision Letter 1]

24 Aug 2021

PONE-D-21-18336R1

Comparative analysis of the unbinding pathways of antiviral drug Indinavir from HIV and HTLV1 proteases by Supervised Molecular Dynamics simulation

PLOS ONE

Dear Dr. Aryapour,

Thank you for submitting your manuscript to PLOS ONE. After careful consideration, we feel that it has merit but does not fully meet PLOS ONE’s publication criteria as it currently stands. Therefore, we invite you to submit a revised version of the manuscript that addresses the points raised during the review process.

As you can see, Reviewer 2 still has many serious concerns that you are requested to address as fully as possible in your revised manuscript and in your rebuttal.

We look forward to receiving your revised manuscript.

Kind regards,

Israel Silman

Academic Editor

PLOS ONE

Reviewers' comments:

Reviewer's Responses to Questions

**Comments to the Author**

1. If the authors have adequately addressed your comments raised in a previous round of review and you feel that this manuscript is now acceptable for publication, you may indicate that here to bypass the “Comments to the Author” section, enter your conflict of interest statement in the “Confidential to Editor” section, and submit your "Accept" recommendation.

Reviewer #1: All comments have been addressed

Reviewer #2: (No Response)

2. Is the manuscript technically sound, and do the data support the conclusions?

Reviewer #1: Partly

Reviewer #2: Partly

3. Has the statistical analysis been performed appropriately and rigorously? 

Reviewer #1: Yes

Reviewer #2: No

4. Have the authors made all data underlying the findings in their manuscript fully available?

Reviewer #1: Yes

Reviewer #2: No

5. Is the manuscript presented in an intelligible fashion and written in standard English?

Reviewer #1: Yes

Reviewer #2: Yes

6. Review Comments to the Author

Reviewer #1: (No Response)

Reviewer #2: The authors have addressed some of the my comments. However, overall I feel that the authors have not done enough improvements to their manuscript so that it can be accepted for publication in PLoS One. Therefore, I would have to recommend rejection.

In the following I will some furthermore comments regarding the manuscript.

I am still not convinced at all with their argument that by performing 50 ps of NVT and NPT equilibration runs after the

frame selection before the 500 ps production runs, they solve the "out-of-equilibration problem" of SuMD. I think this is a point that requires much more consideration and justifications than given in the manuscript

The bottom line is that SuMD can give pathways and mechanisms of ligand unbinding. However, it is clear (see https://pubs.acs.org/doi/abs/10.1021/acs.jcim.9b01094) that SuMD cannot give kinetics or dynamics of ligand unbinding due to procedure of running short simulations and selecting a frame for the next iteration. Pathways and mechanisms of ligand unbinding are certainly of interest so insight obtained with SuMD is valuable. However, I would expect user of any method to be honest of the limitations of the method that they are using. In the current version of the manuscript I feel that the authors are not totally honest with the limitations of SuMD.

Furthermore, I would like to reiterate my comment regarding talking about talking about Free Energy Landscapes in Figures 3 and 7. A free energy landscape is an equilibrium property, it is the logarithm of the probability distribution in the collective variables considered. This probability distribution would tell us that what is the (relative) population of the different states if we would run an infinitely long MD simulations. To estimate a FEL from a finite MD simulations, one would need to observe many transitions between the different metastable states (e.g., bound and unbound state).

I believe that the authors can agree with me what they present in Figures 3 and 7 are strictly not true equilibrium free energy landscapes. As I said in my previous report, this is still a useful way to present the results from SuMD simulations. It should just be acknowledged that these are not true equilibrium free energy landscapes, but rather a useful way to present the results from the SuMD in a CV space.

Another issue that I noticed while reading https://pubs.acs.org/doi/abs/10.1021/acs.jcim.9b01094 better (this is the papers from the developers of SuMD where they apply it to ligand unbinding), is that they SuMD procedure used in the current manuscript is different from the one described on page 2 (1805) in https://pubs.acs.org/doi/abs/10.1021/acs.jcim.9b01094. The procedure described there is much more intricate while the procedure used in the current manuscripts in much simpler. For example, in [DOI:10.1021/acs.jcim.9b01094] they use the idea of a "productive short MD run" that is not used here. I would have expect the authors to address and acknowledge this in the their manuscript.

Regarding the data that the authors have made available on Zenodo. The authors state that "Yes - all data are fully available without restriction" and that "all data is available online at: " ext-link-type="uri" xlink:type="simple">https://zenodo.org/record/5121095#.YPpq470zbIU". However, there the authors only include a Gromacs TPR file for each run, and a Gromcas XTC trajectory. Furthermore, there is no information given on how to use the data. In my opinion this does not fulfill the requirements of PLoS of "Authors are required to make all data underlying the findings described fully available, without restriction, and from the time of publication." There is no way of seeing how the conclusion in the manuscript are reached from the data include on Zenodo. I would ask the authors to greatly expand the data included so that the requirement of making available all data needed to reach the conclusions in the manuscript. In my opinion, this should include initial geometries (e.g., Gromcas GRO file), topology files (Gromacs TOP file), and also the script used to perform the simulations. Furthermore, at the very least there should be a readme file describing how to guide people through the data.

7. PLOS authors have the option to publish the peer review history of their article (what does this mean?). If published, this will include your full peer review and any attached files.

Reviewer #1: No

Reviewer #2: No

---

## [Author Response · Author response to Decision Letter 1]

25 Aug 2021

Reviewer #2: The authors have addressed some of my comments. However, overall, I feel that the authors have not done enough improvements to their manuscript so that it can be accepted for publication in PLoS One. Therefore, I would have to recommend rejection. In the following I will some furthermore comments regarding the manuscript.

Question #1: I am still not convinced at all with their argument that by performing 50 ps of NVT and NPT equilibration runs after the

frame selection before the 500 ps production runs, they solve the "out-of-equilibration problem" of SuMD. I think this is a point that requires much more consideration and justifications than given in the manuscript.

Answer #1: This study did not focus on the justification of the SuMD method. It focuses on the applications of SuMD in uncovering the unbinding pathways of a ligand. We believe enough explanations have been added to the text.

Question #2: The bottom line is that SuMD can give pathways and mechanisms of ligand unbinding. However, it is clear (see https://pubs.acs.org/doi/abs/10.1021/acs.jcim.9b01094) that SuMD cannot give kinetics or dynamics of ligand unbinding due to procedure of running short simulations and selecting a frame for the next iteration. Pathways and mechanisms of ligand unbinding are certainly of interest so insight obtained with SuMD is valuable. However, I would expect user of any method to be honest of the limitations of the method that they are using. In the current version of the manuscript I feel that the authors are not totally honest with the limitations of SuMD.

Answer #2: Apart from the fact that SuMD is an out-of-equilibrium method, it is a great method to achieve a mechanistic and energetic insight into the unbinding pathway of a ligand, and that was the area we focused on in this study. We did not focus on the kinetics parameters of the unbinding pathway. We even mentioned that the duration times of the unbinding pathways of both ligands are in contrast with the real potentials of the ligands: "Although Indinavir inhibits HIV protease much more selectively than the HTLV-1 protease, the overall time needed for the unbinding events of the HTLV-1 protease case was considerably more than that of the HIV protease case. However, many factors govern an inhibitor's activity, and only comparing the overall duration of the unbinding events in three series of replicas is not a correct way of comparing these two cases. This may be due to the inherent limitations of the SuMD method. Because the time window is set to 500 ps, therefore more conformational sampling by ligand is limited."

Question #3: Furthermore, I would like to reiterate my comment regarding talking about talking about Free Energy Landscapes in Figures 3 and 7. A free energy landscape is an equilibrium property, it is the logarithm of the probability distribution in the collective variables considered. This probability distribution would tell us that what is the (relative) population of the different states if we would run an infinitely long MD simulations. To estimate a FEL from a finite MD simulations, one would need to observe many transitions between the different metastable states (e.g., bound and unbound state).

I believe that the authors can agree with me what they present in Figures 3 and 7 are strictly not true equilibrium free energy landscapes. As I said in my previous report, this is still a useful way to present the results from SuMD simulations. It should just be acknowledged that these are not true equilibrium free energy landscapes, but rather a useful way to present the results from the SuMD in a CV space.

Answer #3: The phrase "out-of-equilibrium" was added to the FEL section to clarify this fact for the readers.

Question #4: Another issue that I noticed while reading https://pubs.acs.org/doi/abs/10.1021/acs.jcim.9b01094 better (this is the papers from the developers of SuMD where they apply it to ligand unbinding), is that they SuMD procedure used in the current manuscript is different from the one described on page 2 (1805) in https://pubs.acs.org/doi/abs/10.1021/acs.jcim.9b01094. The procedure described there is much more intricate while the procedure used in the current manuscripts in much simpler. For example, in [DOI:10.1021/acs.jcim.9b01094] they use the idea of a "productive short MD run" that is not used here. I would have expect the authors to address and acknowledge this in the their manuscript.

Answer #4: An explanation about the difference was added to the method section.

Question #5: Regarding the data that the authors have made available on Zenodo. The authors state that "Yes - all data are fully available without restriction" and that "all data is available online at: https://zenodo.org/record/5121095#.YPpq470zbIU". However, there the authors only include a Gromacs TPR file for each run, and a Gromcas XTC trajectory. Furthermore, there is no information given on how to use the data. In my opinion this does not fulfill the requirements of PLoS of "Authors are required to make all data underlying the findings described fully available, without restriction, and from the time of publication." There is no way of seeing how the conclusion in the manuscript are reached from the data include on Zenodo. I would ask the authors to greatly expand the data included so that the requirement of making available all data needed to reach the conclusions in the manuscript. In my opinion, this should include initial geometries (e.g., Gromcas GRO file), topology files (Gromacs TOP file), and also the script used to perform the simulations. Furthermore, at the very least there should be a readme file describing how to guide people through the data.

Answer #5: We provided the TPR and the XTC files of each replica in the Zenodo database. These two files are enough for all of the analysis of this study. You can extract the topology, initial geometries, and much more files only by having these two files. Therefore we believe it completely fulfills the journal requirements.

---

## [Decision Letter · Decision Letter 2]

14 Sep 2021

Comparative analysis of the unbinding pathways of antiviral drug Indinavir from HIV and HTLV1 proteases by Supervised Molecular Dynamics simulation

PONE-D-21-18336R2

Dear Dr. Aryapour,

We’re pleased to inform you that your manuscript has been judged scientifically suitable for publication and will be formally accepted for publication once it meets all outstanding technical requirements.

Kind regards,

Israel Silman

Academic Editor

PLOS ONE

Additional Editor Comments (optional):

Reviewers' comments:

Reviewer's Responses to Questions

**Comments to the Author**

1. If the authors have adequately addressed your comments raised in a previous round of review and you feel that this manuscript is now acceptable for publication, you may indicate that here to bypass the “Comments to the Author” section, enter your conflict of interest statement in the “Confidential to Editor” section, and submit your "Accept" recommendation.

Reviewer #2: (No Response)

2. Is the manuscript technically sound, and do the data support the conclusions?

Reviewer #2: Partly

3. Has the statistical analysis been performed appropriately and rigorously? 

Reviewer #2: No

4. Have the authors made all data underlying the findings in their manuscript fully available?

Reviewer #2: No

5. Is the manuscript presented in an intelligible fashion and written in standard English?

Reviewer #2: Yes

6. Review Comments to the Author

Reviewer #2: I am still not fully convinced by arguments of the authors regarding the SuMD methods. Furthermore, I do not agree with how the authors present the SuMD method in this manuscript. However, I can agree that this is more of a scientific disagreement. Therefore, I can agree that the manuscript is accepted for publication in PLoS One.

As for the data that the authors have deposited on Zenodo and their reply regarding that. I still stand by my previous comment for the revised (R1) version. I feel that the authors are trying to do the bare-minimum to fulfill the data sharing requirement of PLoS, rather than really trying following an open science policy like PLoS one is trying to foster. Unfortunately, this type of attitude is prevalent in the simulation community. Therefore, I still answer "No" to the question "Have the authors made all data underlying the findings in their manuscript fully available?" I leave it up to the editor have to address this issue.

7. PLOS authors have the option to publish the peer review history of their article (what does this mean?). If published, this will include your full peer review and any attached files.

Reviewer #2: No

---

## [Editor Report · Acceptance letter]

16 Sep 2021

PONE-D-21-18336R2 

Comparative analysis of the unbinding pathways of antiviral drug Indinavir from HIV and HTLV1 proteases by supervised molecular dynamics simulation 

Dear Dr. Aryapour:

I'm pleased to inform you that your manuscript has been deemed suitable for publication in PLOS ONE. Congratulations! Your manuscript is now with our production department. 

Kind regards, 

on behalf of

Prof. Israel Silman 

Academic Editor

PLOS ONE